# Glutathione triggers leaf-to-leaf, calcium-based plant defense signaling

Rui Li [1,2,7], Yongfang Yang[1,3,7], Hao Lou[1], Weicheng Wang[1], Ran Du [2], Haidong Chen[2], Xiaoxi Du [1,4], Shuai Hu[1,4], Guo-Liang Wang [5] ✉, Jianbin Yan [2] ✉, Xiaoyi Shan[1] ✉ & Daoxin Xie [1,6] ✉

Animals rely on nervous systems to cope with environmental variability, whereas plants are characterized by lack of nervous system but still have evolved systemic communication systems through signaling molecules that trigger long-distance defense signaling events when encountered with environmental challenges. Here, our genetic screening of the previously constructed hairpin RNA-based *Arabidopsis* library identifies a glutathione (GSH)-deficient mutant that has high accumulation of glutamate (Glu), a previously defined wound signal essential for activating plant defense, but disharmoniously exhibits attenuation of defense signaling events. We further uncover GSH as a critical signaling molecule that relies on GLUTAMATE RECEPTOR-LIKE 3.3 (GLR3.3) to trigger long-distance calcium-based defense signaling events in plants. Our findings offer new insights into highly sophisticated systemic defense systems evolved by plants to defend against herbivory and pathogen invasion.

Animals rely on well-developed nervous systems to dexterously take prompt movements and rapidly facilitate long-range information exchange in front of the changing environment. In mammalian nervous systems, many different types of chemical substances, including amino acids (e.g., glutamate), monoamines (e.g., dopamine, norepinephrine), acetylcholine, peptides (e.g., substance P), serve as neurotransmitters to bind their corresponding receptors, then activate ion channels, and consequently produce ion flux[1], thereby inducing membrane potential changes that lead to signal propagation for neurotransmission[1-4].

Plants are characterized by lack of a nervous system but still have the ability to systematically communicate the environmental signals from the site of perception to the distal tissues[5]. In response to herbivorous insect attack or wounding at local leaves, wound signaling

molecules are released at the damaged plant cells, and may bind corresponding receptors to trigger propagation of slow wave potentials (SWPs)[6-9], initiate transmission of cytoplasmic calcium ([Ca$^{2+}$]$_{cyt}$)[10-15], and consequently activate biosynthesis of defensive phytohormone jasmonic acid (JA), thereby mediating expression of defense-responsive genes for systemic defense responses in plants[7,8,11,12,16-18].

Many different types of chemical substances were revealed as neurotransmitters that induce long-distance ionic responses to effectively mediate neurotransmission in mammals[4]. However, in plant kingdom, only an amino acid and a protein, namely the glutamate (Glu) and the recently discovered "Ricca's factor" β-THIOGLUCOSIDE GLUCOHYDROLASE (TGG) are so far defined as wound signals that depend on glutamate receptor-like (GLR) protein, the homologous protein of mammalian ionotropic Glu receptors (iGluRs), to activate systemic

[1]MOE Key Laboratory of Bioinformatics, Tsinghua-Peking Joint Center for Life Sciences, and School of Life Sciences, Tsinghua University, Beijing 100084, China. [2]Shenzhen Branch, Guangdong Laboratory of Lingnan Modern Agriculture, Key Laboratory of Synthetic Biology, Ministry of Agriculture and Rural Affairs, Agricultural Genomics Institute at Shenzhen, Chinese Academy of Agricultural Sciences, Shenzhen, China. [3]Key Laboratory of Seed Innovation, Institute of Genetics and Developmental Biology, Chinese Academy of Sciences, Beijing 100101, China. [4]Hainan Key Laboratory for Biosafety Monitoring and Molecular Breeding in Off-Season Reproduction Regions, Institute of Tropical Bioscience and Biotechnology & San Ya Research Institute, Chinese Academy of Tropical Agricultural Sciences, Haikou, China. [5]Department of Plant Pathology, Ohio State University, Columbus, OH 43210, USA. [6]State Key Laboratory of Hybrid Rice, Hunan Hybrid Rice Research Center, Hunan Academy of Agricultural Sciences, Changsha 410125, China. [7]These authors contributed equally: Rui Li, Yongfang Yang. ✉e-mail: wang.620@osu.edu; jianbinlab@caas.cn; shanxy80@mail.tsinghua.edu.cn; daoxinlab@mail.tsinghua.edu.cn

plant defense response[6,7,11]. In this study, our genetic screening of the previously constructed hairpin RNA-based *Arabidopsis* library[19] identified a glutathione (GSH)-deficient mutant that highly accumulated Glu, the previously defined molecule crucial for activation of plant defense, but disharmoniously exhibited attenuation of defense signaling events. We further uncovered GSH as a critical signaling molecule that relies on GLR3.3 to trigger long-distance propagation of calcium-based defense signaling events.

## Results

### Identification of GSH as a signaling molecule that activates long-distance plant defense signaling events

During the genetic screening of the rolling circle amplification-mediated hairpin RNA (RMHR)-based *Arabidopsis* library[19,20] for mutants with altered defense response to a disastrous plant fungal pathogen *Botrytis cinerea* (*B. cinerea*), we identified a mutant, termed *hairpin RNA-S2* (*hr-S2*), which exhibited obvious susceptibility to *B. cinerea* infection (Supplementary Fig. 1). A single gene *AT4G23100* was identified with PCR analysis on the mutant genomic DNA[19,20].

To further verify that the decreased resistance in the *hr-S2* mutant is caused by the interference with *AT4G23100* gene, we obtained the previously reported mutant *pad2-1* with the Ser298Asn substitution (S298N) within AT4G23100[21] to assess its responses to *B. cinerea* infection and insect attack. The results showed that *pad2-1* exhibited severe susceptibility to *B. cinerea* and *Spodoptera exigua*, a globally agricultural pest, while genetic transformation of *pad2-1* with *PAD2* (*AT4G23100*) completely restored the resistance (Supplementary Figs. 2 and 3). Our results together with the previous findings[21–23] demonstrate that *AT4G23100* gene is essential for plant defense response against *B. cinerea* infection and insect attack.

It was known that the *AT4G23100* gene encodes the gamma-glutamylcysteine synthetase catalyzing the reaction between Glu and cysteine for generation of glutamylcysteine in the GSH biosynthesis pathway[24]. We found that a high level of Glu (875.42 µg/g FW) was accumulated in the *pad2-1* mutant compared to WT (494.96 µg/g FW) while the GSH content in *pad2-1* (4.91 µg/g FW) was severely decreased to 5.9% of WT (82.94 µg/g FW) (Fig. 1a, b). It is surprising that such a high level of Glu failed to induce plant defense in *pad2-1* (Supplementary Figs. 2 and 3), which seems to be inconsistent with the previously defined role of Glu as a critical wound signal that activates plant defense response[11]. The observed susceptibility in the GSH-deficient *pad2-1* mutant suggests an indispensable role for GSH in plant defense, which encourages us to further investigate whether GSH acts as wound signal to trigger long distance calcium-based defense signaling events.

To verify the role of GSH in activating long-distance plant defense signaling events, we wounded leaf 1 of *pad2-1* and Col-0, and then measured the accumulation of wound-induced plant defensive hormonal molecules in leaf 6, considering that leaf n shares direct vascular connections with leaves n ± 5 and n ± 8 while leaves n ± 3 may represent contact parastichies formed by proximal but unconnected vasculature in adult *Arabidopsis* rosettes[7,25]. Consistent with the reduced plant resistance in *pad2-1*, the defensive phytohormone jasmonates were obviously decreased in the leaf 6 of *pad2-1* to ~63.04% (for JA) and ~45.81% (for JA-Ile) of those in Col-0 upon wounding leaf 1 (Fig. 1c, d). Moreover, the expression of JA-inducible defense marker genes *OPR3*, *JAZ5*, and *JAZ7* was also significantly attenuated in leaf 6 of *pad2-1* compared to Col-0 upon wounding leaf 1 (Fig. 1e). These results suggest that GSH might serve as a critical signaling molecule essential for activation of plant systemic defense signaling events.

To further explore the underlying mechanism of GSH in activating systemic plant defense responses, we introduced the $[Ca^{2+}]_{cyt}$ reporter GCaMP3 into *pad2-1*, generating *GCaMP3/pad2-1* plant, to investigate whether the GSH-deficient mutant attenuated the wound-induced systemic $[Ca^{2+}]_{cyt}$ transmission. Upon mechanical wounding of leaf 1, both amplitude and kinetics of $[Ca^{2+}]_{cyt}$ propagation in leaf 6 were

reduced in *GCaMP3/pad2-1* compared with *GCaMP3* plants (Fig. 1f, g; Supplementary Fig. 4; Supplementary Movies 1 and 2). The peak value of $[Ca^{2+}]_{cyt}$ fluorescence changes in leaf 6 of *GCaMP3/pad2-1* was reduced to ~44.15% of that in *GCaMP3* (Fig. 1h). Consistently, the percentage of each systemic leaf that showed apparent increased $[Ca^{2+}]_{cyt}$ fluorescence was reduced in *GCaMP3/pad2-1* (Fig. 1i). These data reinforce the role of GSH in triggering the wound-induced systemic $[Ca^{2+}]_{cyt}$ propagation in plants.

Taken together, the above-mentioned data suggest that GSH might serve as an important wound signal that triggers wound-induced systemic $[Ca^{2+}]_{cyt}$ transmission, activates JA biosynthesis, and regulates plant defense responses.

### GSH triggers long-distance $[Ca^{2+}]_{cyt}$ transmission

To further validate that GSH acts as a wound signal triggering long-distance plant defense signaling events, we applied GSH (with Glu and sorbitol as positive and negative controls, respectively) to leaf 1 of the *GCaMP3* plants and visualized the systemic $[Ca^{2+}]_{cyt}$ in the whole seedlings (Fig. 2a, b). Intriguingly, the fast and transient increase of $[Ca^{2+}]_{cyt}$ was notably triggered in systemic leaves (including leaf 3, 4, and 6) by GSH (Supplementary Movie 3) at the level comparable to that induced by Glu (Fig. 2a). Specifically, $[Ca^{2+}]_{cyt}$ rapidly increased to the peak value in leaf 6 at ~230 s upon treatment on leaf 1 with GSH rather than sorbitol (Fig. 2c; Supplementary Fig. 4), and more distal leaves showed $[Ca^{2+}]_{cyt}$ propagation with the increased GSH concentrations (Supplementary Fig. 5). Moreover, the GSH-triggered systemic $[Ca^{2+}]_{cyt}$ transmission was dramatically repressed by pretreatment on the petiole of the wounded leaf with the $Ca^{2+}$ channel inhibitor $LaCl_3$ (Fig. 2e-g). To exclude the possibility that GSH is cleaved by enzymes to release glutamate for $Ca^{2+}$ signal propagation, we employed GGsTop, a well-known chemical inhibitor that specifically targets gamma-glutamyl transpeptidase (GGT), the enzyme responsible for converting GSH to Glu[26]. Our results showed that there were no significant differences in the $Ca^{2+}$ signaling response when GSH was used in combination with GGsTop compared to that of GSH treatment alone (Supplementary Fig. 6), suggesting that GSH was not converted to Glu for activation of systemic $Ca^{2+}$ signaling. Additionally, we synthesized another tripeptide (Glu-Pro-Ala), which contains glutamate along with two different amino acids, and investigated its effects in triggering systemic $Ca^{2+}$ signal propagation in plants. Unlike GSH treatment, application of Glu-Pro-Ala failed to induce long-distance transmission of $Ca^{2+}$ signal in plants (Supplementary Fig. 7), reinforcing the role of GSH as a signaling molecule that effectively triggers systemic $[Ca^{2+}]_{cyt}$ propagation in plants.

Consistently, GSH application on leaf 1 significantly induced the expression of defense-responsive genes (*OPR3*, *JAZ5*, and *JAZ10*) in leaf 6 (by ~5.6, ~8.6 and ~5.7 folds), which is comparable to that induced by Glu (Fig. 2d). Moreover, pretreatment on the petiole of the wounded leaf with $LaCl_3$ severely reduced the GSH-inducible expression of defense genes in systemic leaves (Fig. 2h), verifying that propagation of $[Ca^{2+}]_{cyt}$ is required for GSH-induced defense gene expression. Collectively, these results demonstrate that GSH acts as an important wound signal in triggering leaf-to-leaf, calcium-based plant defense signaling.

### GSH activates the transcriptional expression of defense-related genes in plant systemic leaves

To get deeper insights into the GSH-triggered systemic defense signaling events in plants, we performed transcriptome profiling on leaf 6 collected after 1 h application of GSH to leaf 1 of WT plants. We identified 378 differentially expressed genes (DEGs) in the systemic leaf of GSH-treated plants compared to non-stimulated mock control plants (Supplementary Data 1). Among them, 354 genes including a large number of key defense-responsive genes were upregulated while 24 genes were downregulated (Fig. 3a), which is

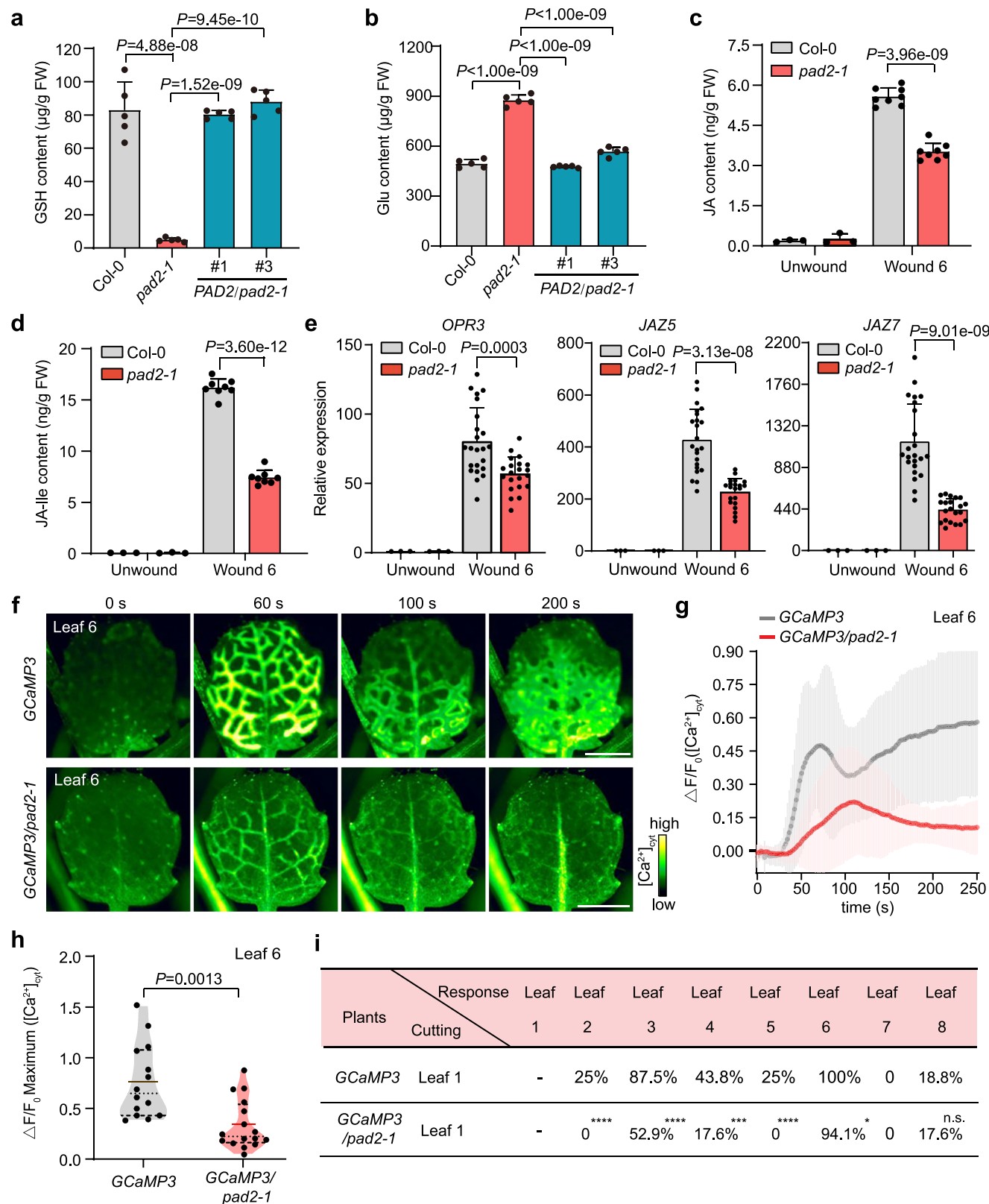

well consistent with the tendency of transcriptome data from plants in response to wounding, namely, more transcripts are upregulated than downregulated[27,28]. Gene Ontology analysis (GO) of these DEGs further revealed significant enrichment of terms associated with plant defense signaling, including defense response, cellular responses to JA, response to wounding, and response to hormone (Fig. 3b; Supplementary Data 2). Consistent with the transcriptome

analysis, qRT-PCR also showed that expression of defense marker genes (*CYP94C1*, *JAZ13*, *LOX4*, *OPR3*, *JAZ5*, *JAZ7*, and *JAZ10*) was highly induced in the systemic leaf tissues of GSH-treated plants (Fig. 3c, d; Supplementary Fig. 8). These data provide an overview of the GSH-induced gene expression in plant systemic leaf tissues and further indicate the important role of GSH in triggering plant long-distance defense signaling.

**Fig. 1 | Defense signaling events are attenuated in GSH-deficient *pad2-1* mutant plant. a, b** Contents of GSH (**a**) and Glu (**b**) in Col-0, *pad2-1*, *PAD2/pad2-1*#1 and *PAD2/pad2-1*#3 plants. Data are mean ± SD ($n = 5$). Statistical significance was determined by Dunnett's test. **c–e** JA (**c**), JA-Ile (**d**) contents and defense gene expression levels (**e**) in target leaf 6 before and 30 min after wounding leaf 1 of Col-0 and *pad2-1*. Data are mean ± SD ($n = 3$ for leaf 6 from unwounded plants), $n = 8$ for leaf 6 from wounded plants in (panel **c**, **d**); $n = 3$ for leaf 6 from unwounded plants, $n = 21$ for wounded *pad2-1* and $n = 23$ for wounded Col-0 plants in (panel **e**). Statistical significance was determined by two-sided Welch's *t*-test. **f** [Ca$^{2+}$]$_{cyt}$ fluorescence signal imaging in target leaf 6 of *GCaMP3* and *GCaMP3/pad2-1* plants upon cutting leaf 1 (0 s). Scale bars: 1 mm. **g, h** Quantitative measurement of [Ca$^{2+}$]$_{cyt}$

levels (**h**) and maximum [Ca$^{2+}$]$_{cyt}$ fluorescence changes (**i**) in target leaf 6 after wounding leaf 1 of *GCaMP3* and *GCaMP3/pad2-1* plants. Data are mean ± SD ($n = 14$ for GCaMP3, $n = 16$ for GCaMP3/pad2-1). Statistical significance was determined by two-sided Welch's *t*-test. **i** Percentage of cases with [Ca$^{2+}$]$_{cyt}$ transmission for each systemic leaf in *GCaMP3* and *GCaMP3/pad2-1* plants wounded at leaf 1. 16 *GCaMP3* and 17 *GCaMP3/pad2-1* plants were subjected to observation of [Ca$^{2+}$]$_{cyt}$ fluorescence and only systemic leaf showing obvious [Ca$^{2+}$]$_{cyt}$ fluorescence was calculated as a case with [Ca$^{2+}$]$_{cyt}$ transmission. Statistical significances compared to *GCaMP3* was assessed using Fisher's exact test, indicated as follows: *$P < 0.05$, ***$P < 0.001$, ****$P < 0.0001$, n.s. denotes not significant. Source data are provided as a Source Data file.

To further investigate the relations between GSH and Glu in plant wound response, we generated transcriptome data sets from leaf 6 1 h after wounding leaf 1 (Supplementary Data 3) or application of Glu to leaf 1 (Supplementary Data 4), and then compared these data to that induced by GSH. This comparison showed that 207 DEGs were co-regulated by wounding and GSH treatment, while 228 DEGs were found in both GSH and Glu treatment groups. Moreover, a total of 267 DEGs from GSH- and Glu-treated groups existed in wounding treatment, among which 126 genes were co-regulated by GSH, Glu and wounding, 81 genes were only regulated by GSH and wounding, and 60 genes were only regulated by Glu and wounding (Fig. 3e). These overlapped genes which are co-regulated by GSH, Glu, and wounding or independently regulated by GSH and wounding, Glu and wounding are all mainly involved in plant defense signaling, including plant response to wounding, stress, JA, and biotic stimulus, regulation of cell communication, and innate immune response, suggesting that GSH and Glu are involved in wound-induced plant systemic defense response.

### GLR3.3 is required for GSH-triggered systemic [Ca$^{2+}$]$_{cyt}$ propagation

GLR proteins which generally contain an amino-terminal domain (ATD), a ligand-binding domain (LBD), a transmembrane domain (TMD), and a carboxyl-terminal domain (CTD), have broad agonist profiles including glycine, cysteine, methionine, Glu and GSH[29–32]. Specially, Glu-triggered long-distance [Ca$^{2+}$]$_{cyt}$ propagation is dramatically inhibited in *glr3.3glr3.6* mutant[11]. To figure out whether GSH-triggered long-distance calcium signaling depends on *GLR3.3* and *GLR3.6*, we compared the GSH-triggered Ca$^{2+}$-related signaling events among the *glr3.3*, *glr3.6* and WT plants containing the GCaMP3 reporter (Fig. 4a, b; Supplementary Fig. 9). Interestingly, the GSH-triggered systemic [Ca$^{2+}$]$_{cyt}$ transmission was severely reduced in the *GCaMP3/glr3.3* (Fig. 4a, b; Supplementary Movie 4). However, the GSH-triggered systemic [Ca$^{2+}$]$_{cyt}$ transmission was not markedly attenuated in *GCaMP3/glr3.6*, which is similar to that of *GCaMP3* plant (Fig. 4a, b; Supplementary Movie 5). Consistently, the GSH-inducible expression of defense marker gene *OPR3*, *JAZ5*, and *JAZ10* was significantly repressed in the *GCaMP3/glr3.3*, but not obviously inhibited in the *GCaMP3/glr3.6* and the *GCaMP3* plants (Fig. 4c). These results suggest that GLR3.3 plays a dominant role in regulating GSH-induced [Ca$^{2+}$]$_{cyt}$ propagation. Similar to the systemic calcium signal transmission upon GSH treatment, Glu-induced [Ca$^{2+}$]$_{cyt}$ propagation in systemic leaves were strongly inhibited in *GCaMP3/glr3.3* while remained unaffected in *GCaMP3/glr3.6* (Supplementary Fig. 10), confirming that Glu-induced long-distance calcium wave is also dependent on GLR3.3[8,11].

To characterize the interaction between GLR3.3 and GSH, the predicted structure of GLR3.3 was used to perform molecular docking. The binding model showed that Glu binds to the LBD of GLR3.3 (referred to as GLR3.3$_{LBD}$) (Supplementary Fig. 11), which is consistent with the recent report about the Glu binding to the GLR3.3$_{LBD}$[29]. While GSH was predicted to bind the ATD of GLR3.3 (referred to as GLR3.3$_{ATD}$), which is remarkably different from Glu in terms of binding domains and contacting amino acids

(Supplementary Fig. 11). Consistent with the predicted interaction between GLR3.3$_{ATD}$ and GSH (Supplementary Fig. 11), recent X-ray crystallography and cryo-EM data revealed that GSH binds the ATD of GLR3.4[30], another *GLR* protein with high sequence similarity to GLR3.3, reinforcing that GSH might bind the ATD domain of GLR3.3 for activation of ion channel. Together with the genetic evidence showing GLR3.3 is required for both GSH- and Glu-triggered systemic [Ca$^{2+}$]$_{cyt}$ transmission (Fig. 4; Supplementary Fig. 10), these results collectively suggest that GSH and Glu may bind to different domains of GLR3.3 for activating ion channel activity and triggering leaf-to-leaf, calcium-based plant defense signaling.

Taken together, our results shown in Figs. 1–4 suggest that GSH serves as a critical wound signal to trigger long-distance propagation of [Ca$^{2+}$]$_{cyt}$ waves in a GLR3.3-dependent manner, thus inducing rapid JA biosynthesis in systemic leaf organs to protect plants from herbivory ahead of time.

## Discussion

Initiation of a rapid defense against biotic attacks and mechanical damages is crucial for plant survival in changing environment. Due to lack of a nervous system, plants have evolved sophisticated mechanisms to systemically deliver the wound signal from the site of injured organs to the distal undamaged tissues, which helps them be well-prepared for upcoming dangers[33]. In this study, we find that the GSH-deficient mutant, in spite of the highly accumulated Glu previously defined as a critical wound signal essential for activating plant defense events, showed impaired plant defense signaling events (Fig. 1; Supplementary Figs. 1-3). We further define GSH as a critical wound signal that depends on GLR3.3 to trigger long-distance calcium-based defense signaling in plants (Figs. 2–4; Supplementary Figs. 5-11). Intriguingly, in the animal nervous system, GSH has been identified as a neuromodulator/neurotransmitter[34–36], suggesting a potential conceptual parallel in GSH-mediated signal transmission across different kingdoms of life. Moreover, we noticed that GSH and Glu induced distinct [Ca$^{2+}$]$_{cyt}$ transmission patterns in systemic leaves. Specifically, the spread of systemic [Ca$^{2+}$]$_{cyt}$ triggered by GSH is preferentially initiated from marginal veins and rapidly transmitted to the mid veins (Supplementary Fig. 12 and Supplementary Movie 6), whereas Glu mediated [Ca$^{2+}$]$_{cyt}$ transmission mainly propagated from mid veins to marginal veins (Supplementary Fig. 12 and Supplementary Movie 7). Re-examination of the data shown in the previous study revealing Glu as a wound signal also supports our observation that Glu triggers [Ca$^{2+}$]$_{cyt}$ transmission with the direction of mid-to-marginal veins[11]. Intriguingly, the transmission pattern of wounding-induced [Ca$^{2+}$]$_{cyt}$ contained both directions of the mid-to-marginal veins and the marginal-to-mid veins (Supplementary Fig. 12 and Supplementary Movie 8). Future work to elucidate the mechanisms and biological significance of different [Ca$^{2+}$]$_{cyt}$ transmission patterns induced by GSH and Glu would help us to further understand the highly sophisticated systemic defense systems evolved by plants to defend against insect attack.

When plants are subjected to insect herbivory or mechanical wounding, they rapidly activate the JA signaling pathway in whole plant

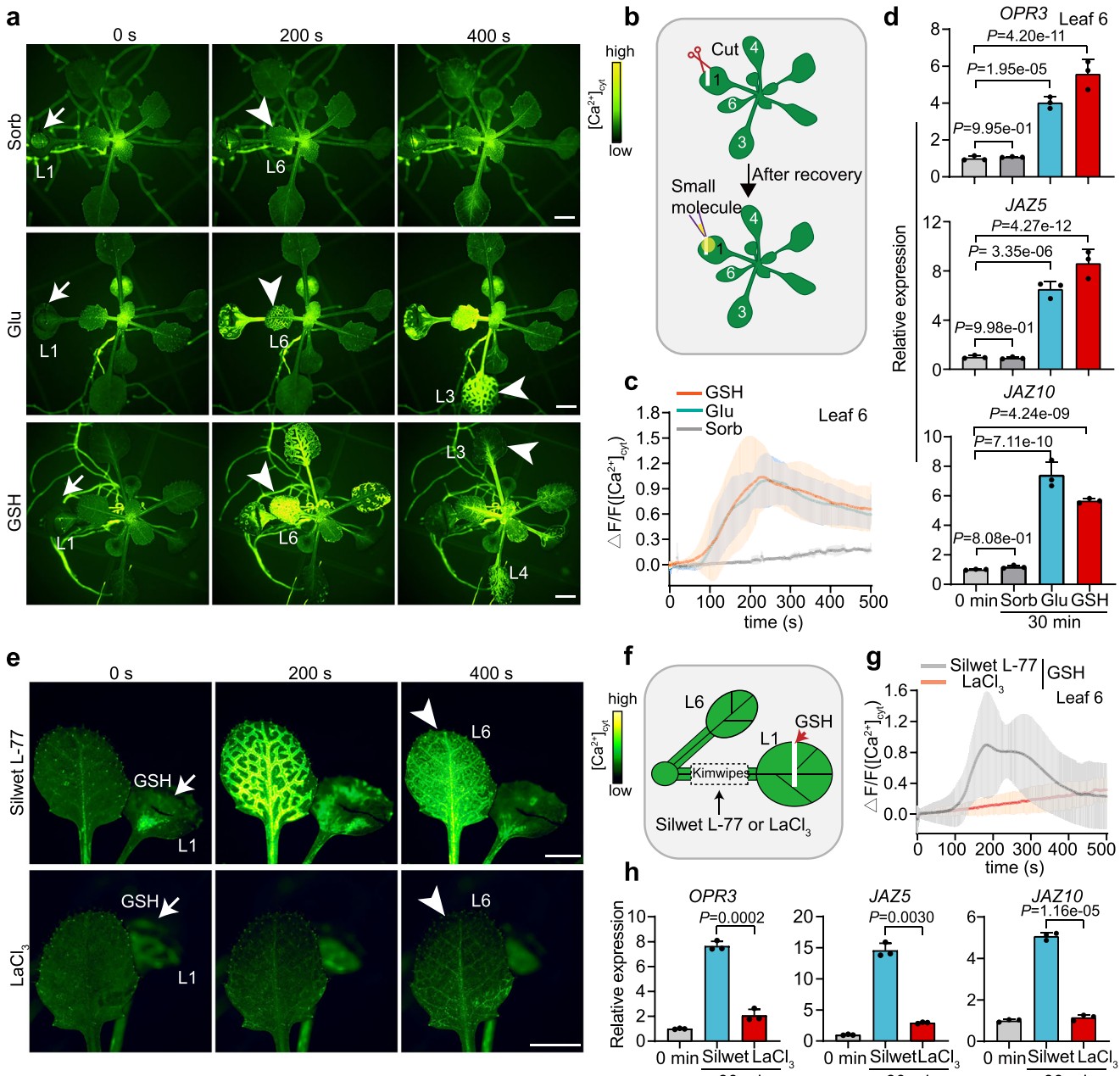

**Fig. 2 | GSH triggers $[Ca^{2+}]_{cyt}$ propagation and defense gene expression in systemic leaves. a** $[Ca^{2+}]_{cyt}$ fluorescence signal imaging of *GCaMP3* plants after 100 mM GSH, Glu, and sorbitol (sorb) application to leaf 1 (L1) (white arrow, 0 s) respectively. White arrowheads (200 s, 400 s) indicate leaf 6 (L6) and leaf 3 (L3). Scale bars: 2 mm. **b** Schematic diagram for application of small molecule to the cut edge of damaged leaf 1. Small molecule was added to the cut edge of the damaged leaf 30 min after wounding for assessment of small molecules-induced $Ca^{2+}$ signal transmission. In contrast, a longer recovery period of 5 h post pre-cutting was required for the assessment of small molecules-activated gene expression levels. **c**, **d** Quantitative measurement of $[Ca^{2+}]_{cyt}$ levels (**c**) and expression levels of defense genes (**d**) in target leaf 6 at indicated times after application of 100 mM GSH, Glu, and sorbitol to leaf 1 of *GCaMP3* plants. Data are mean ± SD ($n = 3, 9, 9$ for sorbitol, GSH and Glu treatment in **c**, n = 3 for d). Statistical significance was determined by Dunnett's test. **e** $[Ca^{2+}]_{cyt}$ fluorescence signal imaging of *GCaMP3* plants pretreated with $LaCl_3$ or Silwet-L77 after 100 mM GSH application to leaf 1 (L1) (white arrow, 0 s). White arrowhead (400 s) indicates leaf 6 (L6). Scale bars: 2 mm. **f** Schematic diagram of $LaCl_3$ pretreatment before GSH treatment. $LaCl_3$ (50 mM) was added to a strip of Kimwipes laid over the petiole of leaf 1 (L1). 0.05% (v/v) Silwet-L77 was added as a wetting agent to improve $LaCl_3$ penetration of the cuticle. **g**, **h** Quantitative measurement of $[Ca^{2+}]_{cyt}$ levels (**g**) and defense gene induction (**h**) in target leaf 6 after application of 100 mM GSH to leaf 1 of *GCaMP3* plant pretreated with $LaCl_3$ or Silwet-L77 at indicated times. Data are mean ± SD ($n = 5$ for g, $n = 3$ for h). Statistical significance was determined by two-sided Welch's *t*-test. Source data are provided as a Source Data file.

tissues, which in turn mediates the systemic defense responses against insects[10–12,37]. In our study, we defined GSH as a critical wound signal to trigger long-distance $[Ca^{2+}]_{cyt}$ transmission for activation of JA biosynthesis in distal organs, thereby inducing plant systemic defense response (Figs. 1–4). Interestingly, as shown in our transcriptome data, in addition to the pronounced enrichment of genes associated with JA

signaling in systemic leaf tissues after GSH treatment, genes responsive to other defense phytohormones such as ethylene and salicylic acid, which are responsible for plant resistance to pathogen infection, are also highly upregulated (Supplementary Data 1), implying that GSH might also act as a signaling molecule to activate the plant systemic immune response. Previously recognized as an antioxidant, GSH is

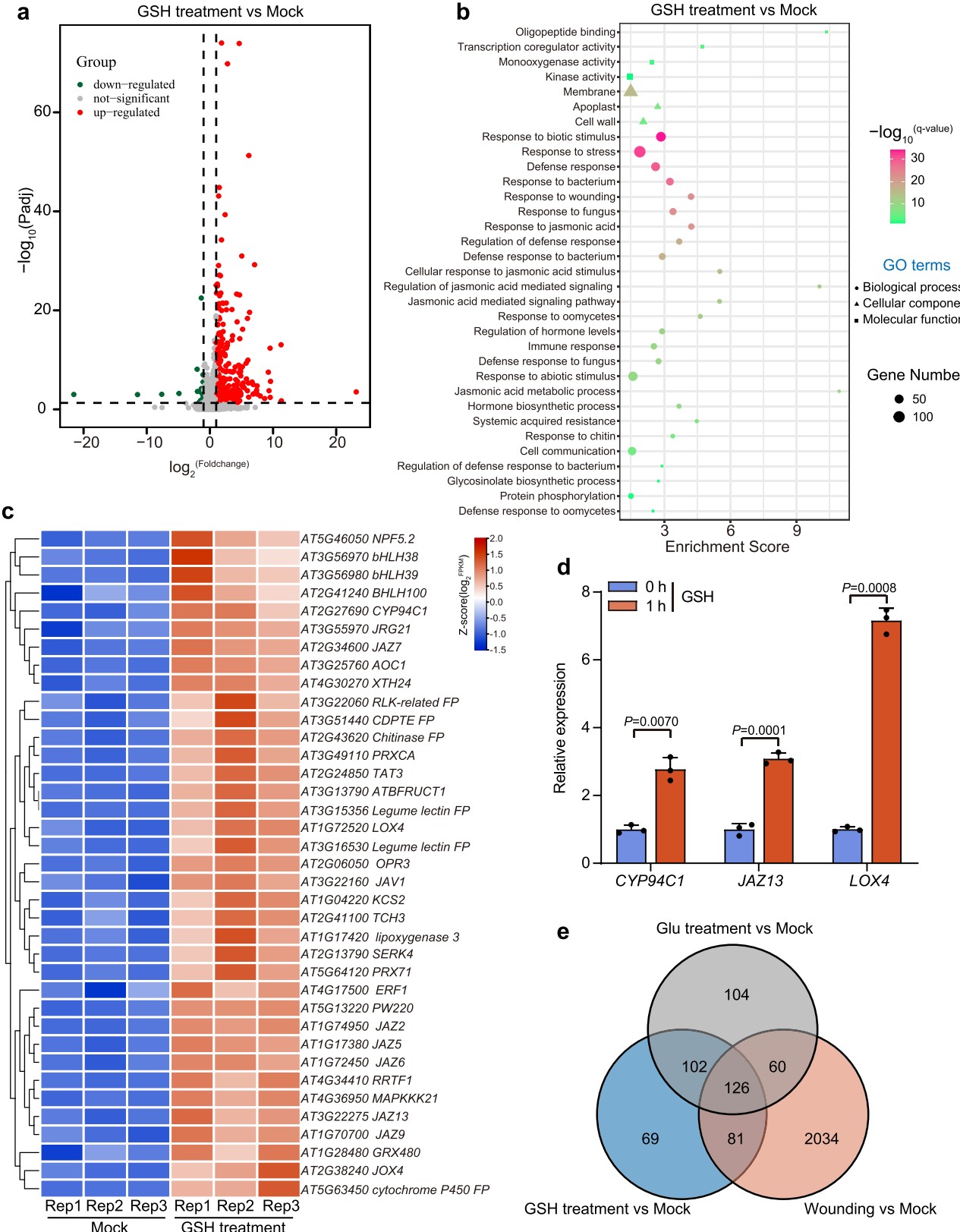

intricately linked with reactive oxygen species (ROS) signaling[38]. Our transcriptome analysis also revealed that genes responsive to oxidative stress were significantly induced in the systemic leaf tissues following GSH treatment (Supplementary Data 1). Given that both calcium and ROS signals play a role in mediating plant systemic defense responses to wounding[11,12,39–41], exploring the potential

involvement of ROS in GSH-triggered systemic defense response would be interesting.

Wound-induced systemic defense signaling requires the involvement of GLR ion channel proteins, such as GLR3.3 and GLR3.6. Our study found that the GSH-triggered long-distance calcium-mediated defense response was suppressed in *GCaMP3/glr3.3* but nearly

**Fig. 3 | Transcriptome analysis of GSH-induced defensive gene expression in plant systemic leaves. a** Volcano plot of differentially expressed genes (DEGs) in target leaf 6 of WT plants after application of 100 mM GSH to leaf 1 compared to non-stimulated mock control plants. **b** Gene ontology (GO) analysis of the responsive genes in target leaf 6 of GSH-treated plants. DEGs in (**a**) were used as input for GO analysis. **c** Heatmap showing the relative expression changes of defense-responsive genes in the target leaf 6 of GSH-treated plants and non-stimulated mock control plants. FP, Family Protein; rep, replicate. **d** The relative expression of selected defense-responsive genes in the target leaf 6 of GSH-treated plants and non-stimulated mock control plants determined by qRT-PCR. Data are mean ± SD ($n = 3$). Statistical significance was determined by two-sided Welch's $t$-test. **e** Venn diagram showing the overlap of DEGs in wounding vs mock, GSH treatment vs mock, and Glu treatment vs mock. Source data are provided as a Source Data file.

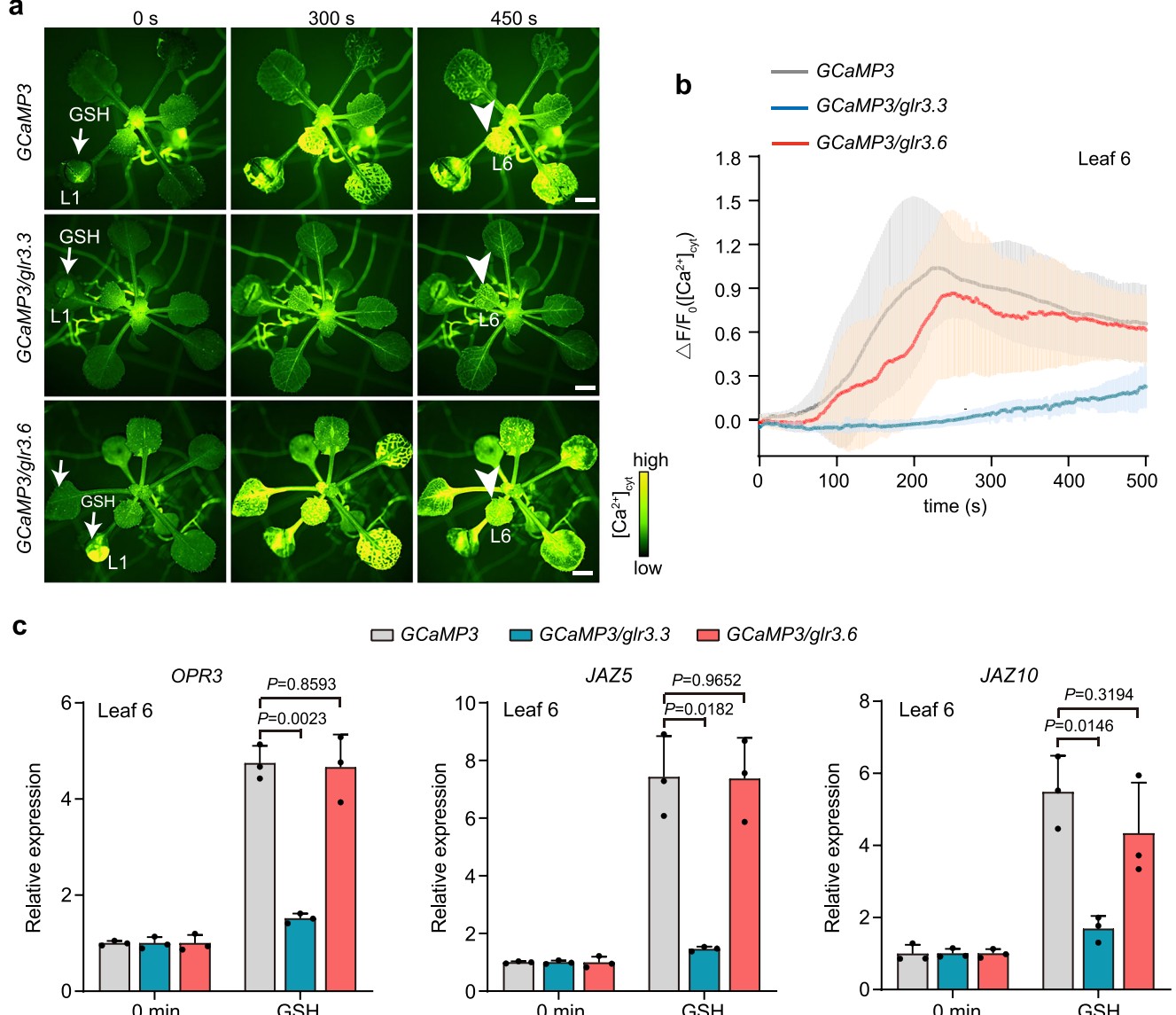

**Fig. 4 | GSH induces systemic [Ca$^{2+}$]$_{cyt}$ propagation and defense gene expression in a GLR3.3-dependent manner. a** [Ca$^{2+}$]$_{cyt}$ fluorescence signal imaging of *GCaMP3*, *GCaMP3/glr3.3* and *GCaMP3/glr3.6* plants after 100 mM GSH application (white arrow, 0 s) to leaf 1 (L1). White arrowhead (450 s) indicates leaf 6 (L6). Scale bars: 2 mm. **b**, **c** Quantitative measurement of [Ca$^{2+}$]$_{cyt}$ levels (**b**) and expression levels of defense genes (**c**) in target leaf 6 at indicated times after 100 mM GSH application to leaf 1 of *GCaMP3*, *GCaMP3/glr3.3* and *GCaMP3/glr3.6* plants respectively. *GCaMP3* data from Fig. 2c are reproduced (gray lines) to aid in comparison. Data are mean ± SD ($n = 5$, 9, 9 for *GCaMP3/glr3.3*, *GCaMP3/glr3.6*, *GCaMP3* in (**b**), $n = 3$ for (**c**)). Statistical significance was determined by two-sided Welch's $t$-test. Source data are provided as a Source Data file.

unaffected in *GCaMP3/glr3.6* (Fig. 4). Similar to GSH, it was also observed that Glu depends on GLR3.3 rather than GLR3.6 to initiate the systemic calcium-based defense signaling (Supplementary Fig. 10). Considering the recent X-ray crystallography and cryo-EM data, which revealed that GSH and Glu could bind to different domains of GLR3.4 for ion channel activation[30], it would be intriguing to investigate whether GSH binds to GLR3.3 and further determine the potential relationship between GSH and Glu in activating GLR3.3. Furthermore, in contrast to GSH and Glu, the recently identified Ricca's factor TGGs, was shown to rely on both GLR3.6 and GLR3.3 to trigger the systemic defense response[6]. It would be exciting to further uncover the signal interactions among Glu, TGGs, and GSH in wound-induced systemic defense signaling. Given that various chemical substances have been identified as neurotransmitters/neuromodulators in mammals,

eliciting long-distance ionic responses that effectively mediate neurotransmission, identification of more systematic signaling molecules in the future would offer new insights into the complex systemic defense signaling in plants.

The successful transmission of neural signals from one cell to another in animals relies on the mutual conversion of chemical signals (neurotransmitters) and electrical signals, where ion fluxes are induced by neurotransmitters and the release of neurotransmitters is dependent on electrical signals[1,2,42]. In contrast, little is known about how signaling substances are transmitted in plants to activate long-distance signaling. Previous studies suggest that the dispersal of elicitors such as Glu may rely on bulk flow and diffusion[14]. Given that GSH has a molecular weight similar to that of amino acids, a plausible scenario is that when plants experience mechanical damage or insect attack, a large volume of GSH from the intracellular region may surge into the apoplastic region, inducing calcium influx by activation of the GLR3.3 ion channel. With the action of diffusion and bulk flow of GSH within the vascular bundles, calcium signals can be rapidly transmitted between leaves. In addition to the diffusion and bulk flow theory, other theories have also been proposed, such as the osmoelectric siphon model, which is based on membrane depolarization leading to cell water shedding into the apoplast followed by membrane repolarization and water uptake[43]. More recently, osmotic pressure has been suggested to induce the systemic apoplastic release of signaling molecules like Glu[15]. Future studies aimed at uncovering the potential mechanisms underlying GSH-triggered systemic $Ca^{2+}$ signal transmission would be fascinating.

## Methods

### Plant materials and growth condition

*hairpin RNA-S2 (hr-S2)* was obtained from the rolling circle amplification-mediated hairpin RNA (RMHR)-based *Arabidopsis* library in our previous study[19,20]. *pad2-1* was ordered from the Nottingham Arabidopsis Stock Centre (NASC). Seeds of *GCaMP3* were from Simon Gilroy (Department of Botany, University of Wisconsin). The *GCaMP3/pad2-1* was generated by genetic crosses using standard procedures. *GCaMP3/glr3.3* and *GCaMP3/glr3.6* mutants were generated by CRISPR/Cas9 in WT plants overexpressing *GCaMP3*. To generate the transgenic plant expressing *35S::PAD2-6myc/pad2-1* (*PAD2/pad2-1*), the coding sequence of *PAD2* (AT4G23100) was cloned into the binary vector *pCAMBIA1300*, and then transformed into *pad2-1*.

*Arabidopsis* seeds were sterilized with 20% bleach, sown on agar plates containing Murashige and Skoog (MS) medium. After incubation in the dark at 4 °C for 2 days, plates were cultivated in a growth chamber at 22 °C with a 16 h/8 h light/dark (long day, LD) photoperiod. After 7 days, seedlings were then transplanted into soil and grown under the same condition. For insect feeding and pathogen inoculation assays, plants were grown in soil for 4–5 weeks at 22 °C with a 10 h/14 h light/dark (short day, SD) photoperiod. For real-time calcium imaging assay, plates were incubated at 4 °C for 2 days in the dark and then transferred to 22 °C with a SD photoperiod for 6–7 days. Seedlings were then transferred onto new plates and grown for additional 14–16 days prior to use.

### Real-time $[Ca^{2+}]_{cyt}$ imaging in the entire plant

Transgenic *Arabidopsis* plants stably expressing *GCaMP3* were imaged with a motorized epifluorescence (SZX10, Olympus) stereomicroscope equipped with a 1 × objective lens and a C13440 digital camera (ORCA-Flash4.0 V3, Hamamatsu Photonics). The GFP-based $Ca^{2+}$ indicator *GCaMP3* was excited using mercury lamp (X-Cite 120 Q, Excelitas). The green fluorescent signal was acquired every 2 s with the C13440 camera using Standard imaging software (Olympus).

Using ImageJ software, GCaMP3 signals were analyzed overtime at regions of interest (ROI). The equation $\Delta F/F_0 = (F-F_0)/F_0$ was used to calculate the fractional fluorescence changes. $F_0$ denotes the average baseline fluorescence determined by the average of F over the frames of the recording before treatment. The entire leaf area was selected as ROI for $Ca^{2+}$ signal analysis.

### Application of amino acids, tripeptides and chemical agents

GSH, Glu-Pro-Ala, L-Glu, and sorbitol were dissolved in growth medium [1/2 × MS salts, 1% (w/v) sucrose and 0.05% (w/v) MES], which was then adjusted to pH 5.1. 5 μL of the solution was applied to the cut edge of leaf 1 after 30 min recovery period after pre-cutting for assessment of small molecules-induced $Ca^{2+}$ signal transmission. In contrast, a longer recovery period of 5 h post pre-cutting was required for the assessment of small molecules-activated gene expression levels. GSH at concentrations of 0.001 mM, 0.01 mM, 0.1 mM, 1 mM, 5 mM, 10 mM, 25 mM, 50 mM, and 100 mM, which are within or exceeding its physiological concentrations, were selected for analyzing GSH-triggered systemic calcium-based defense signaling in vitro. GGsTOP (HY-108467, MCE) was dissolved in DMSO and diluted with water. $LaCl_3$ was dissolved into 0.05% (v/v) Silwet L-77 in water to make a 50 mM solution and then 10 μL was applied to a piece of Kiwmwipes placed on a petiole for at least 30 min prior to adding GSH.

### CRISPR/Cas9 vector construction and plant transformation

Egg cell-specific promoter-controlled CRISPR/Cas9 system was used to generate *glr3.3* and *glr3.6* single mutants. CRISPR/Cas9 vector was constructed according to previous studies[44]. Target sequence was selected by *CRISPR-PLANT* (https://www.genome.arizona.edu/crispr/). All the vectors were transformed into *Agrobacterium tumefaciens* strain GV3101 using electroporation. *Arabidopsis* plants were transformed using the floral dip methods. The primers used for cloning are shown in Supplementary Data 5.

### DNA extraction and mutation detection

1–5 mg frozen leaf tissues were used for DNA extraction through SDS method. The extracted genomic DNA was then used as a template to amplify the desired fragments in each of the target genes using primers flanking the target sites. PCR products were directly sequenced using Sanger method to identify mutation. Superimposed sequence chromatograms produced by biallelic and heterozygous mutations were decoded by DSDecode (http://dsdecode.scgene.com/) and manual analyses.

### Total RNA isolation, cDNA synthesis, and qRT-PCR

TransZol RNA kit (ET101-01, TransGen Biotech) was used to extract total RNA from *Arabidopsis* plants according to manufacturer's instructions. cDNA was synthesized from the total RNA in a 20 μL reaction by cDNA Synthesis SuperMix kit (AT311-03, TransGen Biotech). qRT-PCR was performed in QuantStudio 3 realtime PCR system (Applied Biosystems, USA) using ChamQ SYBR qPCR Master Mix (Low ROX Premixed) (Q331-02, Vazyme). The $2^{-\Delta\Delta Ct}$ analysis method was used to calculate the relative expression levels of each sample.

### JA and JA-Ile quantification in leaf tissues

Metabolite extraction and quantification of JA and JA-Ile in leaf tissues were performed according to previously described methods using AB Sciex 4500 QTRAP triple quadrupole mass spectrometer (AB SCIEX, MA, USA) equipped with an ACQUITY UPLC™ BEH C18 column (Waters, Eschborn, Germany) (50 × 2.1 mm, 1.7 μm)[45]. $d_5$-JA was added to the extraction solvent (isopropanol: formic acid at 99.5: 0.5, v/v) as internal standard. Multiple reaction monitoring (MRM) was applied in negative mode with mass transitions (precursor ions/product) as followed: 209.12 > 59.01 for JA, 322.20 > 130.09 for JA-Ile, and 214.15 > 62.03 for $d_5$-JA. A 10 min gradient elution program was applied for target compounds separation with 0.1% formic acid in acetonitrile (A) and pure water (B) as mobile phase with a 200 μL/min flow rate. Then, mass spectra data were processed and calculated using AB

SCIEX analyst software (version 1.6.3, AB SCIEX, MA, USA) with calculation curves made with standards.

## Measurement of endogenous Glu and GSH in *Arabidopsis*

Leaf tissues were collected and directly ground into powder in liquid nitrogen. Then, 100 mg of powder was weighed accurately and transferred into a 2 mL centrifuge tube. 10 μL of stable isotope internal standard (5 μg/mL $^{13}C_2$-$^{15}N$-GSH, Cambridge Isotope Laboratories) was added to each sample before extraction. Then, 1.5 mL of extraction buffer (80% methanol with 0.1% formic acid, v/v) was added, with vortexing for sample resuspension. After a 15 min centrifugation at $14000 \times g$, the supernatants were diluted 10 times, and 100 μL was transferred to sample vials for further detection.

Target compounds were detected with a 6500 plus QTrap mass spectrometer (AB SCIEX, USA) coupled with a ACQUITY UPLC H-Class system, equipped with a heated electrospray ionization (HESI) probe. Extracts were separated by a synergi Hydro-RP column (2.0 × 100 mm, 2.5 μm, phenomenex). A binary solvent system was used, in which mobile phase A consisted of 2 mM Triisobutylamine adjusted with 5 mM acetic acid in 100% water, and mobile phase B of 100% methanol. An 8 min gradient with flow rate of 500 μL/min was used as follows: 0–1.0 min at 2% B; 1.0-6 min, 5–40% B; 6–7 min, 98% B; 7.1-8 min, 2% B. Column chamber and sample tray were held at 35 °C and 10 °C, respectively. Data were acquired in multiple reaction monitor (MRM) mode with mass transitions (precursor ions/product) as followed: 308.1 > 179.1 for GSH, 147.9 > 83.9 for Glu and 311.0 > 181.9 for $^{13}C_2$-$^{15}N$-GSH. The nebulizer gas (Gas1), heater gas (Gas2), and curtain gas were set at 55, 55, and 35 psi, respectively. The ion spray voltage was 4500 V in positive mode. The optimal probe temperature was determined to be 500 °C. Calibration curves of Glu and GSH were made with standards, and the concentration of GSH was calculated by internal-standard method with $^{13}C_2$-$^{15}N$-GSH as internal standard, while the concentration of Glu was calculated by external-standard method. The SCIEX OS software (version 1.6, AB SCIEX, MA, USA) was applied for metabolite identification and concentration calculation.

## Insect feeding assay with *Spodoptera exigua*

*S. exigua* larvae (2nd instar) were purchased from Jiyuan Baiyun Industry Co., Ltd (China). For feeding assays, 2nd instar *S. exigua* larvae were first underwent 24 h of starvation treatment in growth chamber and were then gently placed on 5 week-old plants. Two *S. exigua* larvae were placed on a plant. Larval weights were measured 7 days after feeding. Representative *S. exigua* larvae recovered from corresponding plants were killed by 75% ethanol solution and photographed by stereomicroscope (SZX10, Olympus) equipped with a 0.75 × objective lens and a digital camera (SC180, Olympus).

## *Botrytis cinerea* infection assay

Conidiospores were collected from *B. cinerea* (B0510) plates and diluted to ~1 × 10⁶ conidiospores/mL in PDB medium (Potato Dextrose Broth, BD). For drop-inoculation assay, 5 μL *B. cinerea* spores was dropped on the center of each detached leaf. Subsequently the detached leaves were placed in dark with high humidity for 2–3 days. Phenotypes were imaged with a digital camera and the lesion sizes were measured using Digimizer software (v3.1.2.0, Belgium, Germany). For spray-inoculation assay, 4 week-old soil-grown seedlings were sprayed evenly with *B. cinerea* conidiospores and kept in high humidity for 2–3 days. The disease severity was classified into four grades: healthy (H, green), light symptoms (L, pale green), severe symptoms (S, orange-yellow), or completely dead (D, red). PDB buffer served as mock in these infection assays. As described in previous studies[45–47], the assessment of *B. cinerea* biomass was determined by real-time PCR quantification of the fungal *CUTINASE* signal (Z69264) in relation to *Arabidopsis ACTIN2*.

## RNA sequencing and data analysis

RNA integrity was checked using the RNA Nano 6000 Assay Kit of the Bioanalyzer 2100 system (Agilent Technologies, CA, USA). Sequencing libraries were generated using NEBNext® UltraTM RNA Library Prep Kit for Illumina® (NEB, USA) following manufacturer's recommendations and index codes were added to attribute sequences to each sample. The clustering of the index-coded samples was performed on a cBot Cluster Generation System using TruSeq PE Cluster Kit v3-cBot-HS (Illumia) according to the manufacturer's instructions. After cluster generation, the library preparations were sequenced on an Illumina Novaseq platform and 150 bp paired-end reads were generated. Low-quality reads were removed from the raw reads using Cutadapt[48] and Trimmomatic[49] software to get clean reads. Clean reads were mapped to the corresponding reference genome using HISAT2 software[50]. Gene expression levels and read count of each gene were calculated using StringTie software[51]. The R package DEseq2 (v1.16.1) was used to identify the differentially expressed genes based on the following criteria: $p_{adj} < 0.05$ and $|\log_2^{FoldChange}| > 1$[52]. Venn diagramming, GO enrichment analysis was performed using TBtools (v2.007)[53].

## Structure prediction and molecular docking

The ab initio structure predictions of AtGLR3.3 protein were implemented by RoseTTAFold and AlphaFold 2 through ColabFold[54–56]. The homology modelling of AtGLR3.3 was conducted using SWISS-MODEL with AtGLR3.4 crystallography model (7lzh) template[30,57]. Ligand mol2 files of GSH and Glu were generated by Avogadro 1.2.0[58]. Prepdock website (https://swift.cmbi.umcn.nl/servers/html/prepdock.html) and AutoDockTools 1.5.7 software were used for adding hydrogens and symmetry related waters, computing Gasteiger charges, and detecting torsion root[59]. Molecular dockings were performed on AutoDock Vina 1.2.0 software and CB-DOCK2 website[60,61]. For AutoDock Vina, the grid box was setting up with a range spacing of 0.897 Å, number of points in x, y and z dimensions as 52, 54, and 70 respectively and center grid box for x, y and z as 173.571, 198.031, and 203.573 respectively for the ATD of AtGLR3.3. Docking of LBD was conducted with 0.756 Å in range spacing. The number of points in x, y and z dimensions was 60, 56, and 78. The center grid box was 172.151, 160.102, and 157.178 in xyz-coordinates. The results were visualized and the polar contacts between ligands and their receptor were detected on PyMOL 2.5.8 software. Hydrogen bonds were determined on UCSF ChimeraX 1.7.1 software with distance tolerance of 0.400 Å and angle tolerance of 20.000°[62].

## Statistical analysis

All data were statistically analyzed using GraphPad Prism 8.0, SPSS, and R software and presented by Adobe Illustrator. Means and deviations were reported in the Figures and the corresponding figure legends.

## Reporting summary

Further information on research design is available in the Nature Portfolio Reporting Summary linked to this article.

# Data availability

The RNA-seq data generated in this study has been deposited in the Gene Expression Omnibus (GEO) database at NCBI under accession code GSE249592. All other data supporting the findings of this study are available in the main text or the Supplementary Data. Source data are provided with this paper.

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

## Acknowledgements

We thank Simon Gilroy (Department of Botany, University of Wisconsin) for providing us with the *GCaMP3* seeds. No conflict of interest is declared. This research was supported by grants from the National Natural Science Foundation of China (32388101 and 32250001 to D.X., 32425011 and 32488302 to J.Y.), National Key R&D Program of China (2022YFD1400800 and 2021YFA1300400 to D.X.), China Postdoctoral Science Foundation (2020M670286 to Y.Y.), Postdoctoral Fellowship of Tsinghua-Peking Joint Center for Life Sciences. The Innovation Program of Chinese Academy of Agricultural Sciences (CAAS) and the Elite Young Scientists Program of CAAS, the New Cornerstone Science Foundation through the XPLORER PRIZE.

## Author contributions

D.X., R.L., Y.Y., Y.J., X.S., G.-L.W. designed the research and experimental strategy. R.L. and Y.Y. performed most of the experiments with the help from H.L., W.W., and S.H. R.D. measured GSH, Glu, JA and JA-Ile contents in plants. H.C. conducted the bioinformatic analysis. X.D. performed molecular docking analysis. Y.Y. and R.L. analyzed the data. R.L. and Y.Y. wrote the manuscript. D.X., X.S., J.Y., R.L. and Y.Y. revised the manuscript.

## Competing interests

The authors declare no competing interests.
