## [Transparent Peer Review file · Nature Communications]

Glutathione triggers leaf-to-leaf, calcium-based plant defense signaling

Corresponding Author: Professor Daoxin Xie

Version 0:

Reviewer comments:

Reviewer #1

(Remarks to the Author)

Wounding signals are transmitted by certain molecules to trigger systemic calcium mediated defense signaling in plants. Previous studies identified glutamate (Glu) and beta-thioglucoside glucohydrolase (TGG) as these molecules. In this study, the authors propose glutathione (GSH) as a new transmitter. They analyzed Arabidopsis thaliana pad2 mutant, which lacks gamma-glutamylcysteine synthetase activity that catalyzes Glu and cysteine for the subsequent generation of GSH. In the mutant, the Glu amount was doubled, and the GSH amount was reduced by 16 times. The authors found that amplitude and kinetics of cytosolic Ca²⁺ levels were reduced in the mutant, and transcript levels of defense response genes downstream of the Ca²⁺ change were diminished. Additionally, exogenous addition of GSH induced the Ca²⁺ wave and the induction of defense genes. Moreover, the Ca²⁺ wave induced by the exogenous GSH addition was diminished in the glutamate receptor glr3.3 mutant. The finding of a new transmitter to regulate defense signaling with Ca²⁺ is influential for a wide community of researchers studying signal transduction, as the function of ligand-gated GLR channels in long-distance Ca²⁺ systemic signaling in plants remains unclear. The text is concise and clear, and the figures are of high quality. The mutant phenotypes and the effects of experimental treatments were well evaluated using multiple approaches, such as Ca²⁺ imaging, detection of marker gene expression, and quantification of metabolites with mass spectrometry. The effort and potential significance of the work presented are appreciated. However, there are some concerns regarding the identification of GSH as a transmitting molecule, and the authors are recommended to conduct additional experiments to verify their findings.

Major concerns:

1. There is concern that the increase in H⁺, rather than GSH, might have induced the Ca²⁺ signaling. 0.05% (w/v) MES in the growth medium (pH 5.1) used to dissolve GSH approximately corresponds to 2.6 mM MES. When calculated using the Henderson-Hasselbalch equation with a pKa of 6.15 for MES, about 90% of 2.6 mM MES is protonated in the growth medium after adjustment to pH 5.1, and the remaining 0.26 mM MES is effective. When GSH is dissolved to 100 mM in the growth medium, 0.26 mM MES concentration is too low to prevent acidification against 100 mM GSH, and the pH value should have decreased to pH3 or less.
2. Further experiments are recommended to exclude the possibility that Glu, rather than GSH, induces the Ca²⁺ signal and defense gene expression. Even though Glu amounts globally increased in pad2 mutant plant body, Glu may further increase locally and function to induce Ca²⁺ increase.
3. The authors speculate that GSH activates GLR3.3 to increase cytosolic Ca²⁺ based on previous reports on the binding of GSH to GLR3.3. However, to identify GSH as a transmitter to Ca²⁺ signal via GLR3.3, it is necessary to examine the activation of GLR3.3 by GSH in vitro.

(minor concerns)

1. line 107: The authors need to deny the possibility that cells are acclimated to increased Glu and becomes less sensitive to wound induced Glu, rather than the decrease of GSH.
2. line. 105: Ca²⁺ measurement of the pad2 mutant with exogenous GSH may support the role of GSH.
3. line 117 (Fig. 2c): Please explain the reason for the difference in graph shapes (3 clear peaks for Glu but 3 gradual peaks

for GSH).

4. line 123: Results of JAZ10 in Fig. 1 and JAZ5 in Fig. 2h should be added for comparison to other figures.
5. line 143: For comparison with Fig. 1 and 2, qRT-PCR of OPR3, JAZ5, and JAZ10 should be added.
6. line 172: JAZ5 and JAZ10 should be analyzed in concordance with other experiments in Figs. 1 and 2.
7. Figs. 1h, 2g, and 4b: ROIs should be indicated.
8. The authors should use Welch's t-test instead of the Student's t-test. The Student's t-test assumes equal variance of the two populations compared, but Welch's t-test does not. It is usually unclear whether the two populations compared have equal variance, so Welch's t-test is more reliable.
9. For multiple comparisons, the authors sequentially use one-way ANOVA and then Dunnett's test. Although such a sequential application of ANOVA and a multiple comparison method is often used, this is logically flawed [Ruxton and Beauchamp (2008) Behavioral Ecology, 19: 690–693]. It is better to simply use Dunnett's test without the prior use of ANOVA.
10. Fig. 1j and Extended Fig. 2e lack statistical tests. For Extended Fig. 2e, Steel's test (non-parametric version of Dunnett's test) can be applicable. For Fig. 1j, Fisher's exact test is one possible choice.

Reviewer #2

(Remarks to the Author)

Reviewer #3

(Remarks to the Author)

The manuscript entitled: Glutathione triggers leaf-to-leaf, calcium-based plant defense signalling submitted by Li et al. to Nature Communications investigates the role of glutathione (GSH) in plant systemic signalling. In their experiments, they use series of elegant genetic and pharmacological approaches to show that GSH is an important component of wound induced Ca²⁺ systemic signalling in plants. So far, this function was ascribed mainly to amino acid L-glutamate and so called Ricca factor, but it has been clear that systemic wound signal is much more complex. Using Arabidopsis intracellular calcium [Ca²⁺]cyt reporter lines and their crosses with glr and pad2-1 mutants, they elegantly demonstrated propagation of [Ca²⁺]cyt signal. Further, they showed that glutathione Ca²⁺ signalling response is dependent on GLR3.3 receptor, similar to L-Glu. Using molecular docking, they suggested a possible binding pocket in ATD subunit of GLR3.3. The study seems to be carefully done with presented experimental setup (but see my comments below), with sufficient repetition and the manuscript is clearly and concisely written.

Major comments:

- Glutathione (GSH) is tripeptide composed of glutamate, glycine and cysteine. You suggested its specific role in systemic Ca²⁺ signal propagation. However, is it possible that after wounding, when cellular content an enzyme compartmentalization is disrupted, the GSH is cleaved and the effect is mediated through released glutamate? Is the effect of GSH indeed a separate phenomenon? GSH induced exactly the same response as L-Glu and both are dependent on GLR3.3. Maybe, more appropriate negative control than osmotic control sorbitol, would be another tripeptide containing glutamate but two other amino acids.
- Transcriptomic analysis (Fig. 3). You applied GSH and L-Glu after 30 min recovery period after leaf pre-cutting. Then, transcriptomic analyses were done 1 hour after application of GSH, L-Glu and wounding and compared to mock control. Does the mock control contain this pre-cutting stimulus? Otherwise, increased expression of defence – related genes in L-Glu and GSH treated plants may be the result of pre-cutting stimulus (analysed 1,5 h after). Recently, Bellandi et al. 2022 (Sci. Adv.) criticized this pre-cutting approach and found that glutamate alone without wounding component cannot induce JA response in systemic leaves. In your movies, it is clearly visible wound after pre-cut treatment and I cannot believe that this intervention does not affect defence-gene expression after 1.5 hours.
- Did you analysed if your data have equal variances and normal distribution? Equal variances and normality are assumptions for t-test and ANOVA.
- Page 5/line107: „Taken together, the above-mentioned data suggest that GSH might serve as an indispensable wound signal that triggers wound-induced systemic [Ca²⁺]cyt transmission, activates JA biosynthesis, and regulates plant defense responses.“ Because the Ca²⁺ signalling and defence response is only attenuated but not completely blocked in pad2-1 mutant, I found this not as indispensable but maybe redundantly dependent. GSH is probably only modulator. Except its role in Ca²⁺ signalling you showed, it is known that GSH can modulate phytohormone response dependent on repressor stabilization/degradation (e.g. IAA and JA signalling), and can act as IAA or JA signalling enhancer, Pasternak et al., 2020). So the increased susceptibility of pad2-1 mutant plants to herbivore damage can be connected also with other mechanisms of GSH action.
- Maybe, you should discuss how the GSH may bind and activate GLR3.3. receptor. A possible scenario is that GSH, as the main transport form of sulphur in phloem, is released upon wounding?

Reviewer #4

(Remarks to the Author)

Li et al. performed a hairpin RNA-based screen in *Arabidopsis* and identified a glutathione (GSH)-deficient mutant in the At4g23100 gene (gamma-glutamylcysteine synthetase, also known as pad2-1). This mutant exhibited a 1.8-fold increase in glutamate levels and a 17-fold decrease in GSH compared to wild-type plants. Hormone assays revealed that pad2-1 mutants had reduced JA-derivatives and lower expression of JA-induced genes upon wounding. Using GCaMP3-based calcium imaging, they observed a diminished Ca²⁺ response to wounding in the mutant. When GSH was applied to GCaMP3 plants, it induced dose-dependent, LaCl₃-sensitive, systemic Ca²⁺ signals and upregulated defense-related genes. Further gene expression analysis highlighted specific clusters of genes differentially regulated by GSH versus glutamate or wounding, many associated with plant defense, JA response, and wounding. Molecular docking studies suggested that GSH might bind to GLR3.3's ATD. Based on their data the authors propose that GSH could serve as a wound signal, initiating systemic Ca²⁺ signaling, JA biosynthesis, and defense responses.

The manuscript is well-written, and the experiments are robust, carefully conducted, and largely presented effectively. The identification of GSH as a potential signaling molecule in systemic wound signaling is a significant discovery with considerable implications for the field. However, the analysis of certain datasets, particularly those related to calcium signaling, along with their discussion and interpretation, require further refinement. A functional model integrating the findings with current literature would be desirable.

Major points to address:

The authors should explicitly state that the GSH concentration used in their experiments is several orders of magnitude higher than the typical cellular concentrations, which reportedly range from 0.01 to 15 mM. While local GSH application (100 mM) to a leaf triggers a systemic calcium response, the kinetics and amplitude of this response differ from those triggered by glutamate. Notably, the GSH-induced systemic Ca²⁺ signal appears to originate at the leaf margin, potentially in secondary or tertiary veins, and propagates inward toward the center. This contrasts with the expected signal pattern from wounding or glutamate application, where the Ca²⁺ wave typically starts at the petiole or main vasculature and moves outward toward the leaf rim (compare Supplemental Movies 3 and 2). Was this pattern consistently observed?

If so, this phenomenon should be emphasized and discussed, as it deviates from previously reported Ca²⁺ signaling patterns induced by glutamate or wounding. If the pattern holds true, it could indicate a spatially distinct response mechanism for GSH. Given these potential spatial differences in calcium signal propagation, it's crucial to ensure that the selection and positioning of the ROIs are comparable for consistent data analysis. Was a specific leaf area or the petiole selected for analysis, or was the entire leaf considered? Please include this methodological detail in the methods section, as it is currently missing.

Was the effect of GSH on the [Ca²⁺]_{cyt} also analysed in the GCaMP3/pad2-1 background? If so, how was the calcium signal compared to the other lines?

The authors should discuss the potential mechanisms underlying GSH-mediated signaling in their analysis and maybe summarise this in a functional model. Given that the largest GSH reserves are intracellular, for GSH to interact with the ATD of GLR3.3s, which are presumed to function on the plasma membrane, it would first need to reach the apoplast.

The authors should comment whether they believe that GSH itself is transported from the wounded site to systemic leaves? If this is the case, how do they reconcile this with the presumably rapid breakdown (within minutes) of GSH in the apoplast, i.e., by enzymes like gamma-glutamyl transpeptidase, which would convert GSH into glutamate? This raises the question of whether glutamate could actually be the "bioactive" signal responsible for the observed effects, at least partially.

To test this hypothesis, the authors could (1) employ a genetically encoded redox sensor like roGFP2 to monitor changes in redox state or, as the function of this sensor might be affected in the apoplast, (2) compare the effects of GSH and glutamate application on plants expressing a genetically encoded glutamate sensor, such as GluSnFR or FLIPE. Since these sensors specifically detect glutamate, an increase in GSH concentration should not affect fluorescence, allowing for a clearer distinction between the roles of GSH and glutamate in signaling.

If the authors suggest that GSH is immobile, they should address what triggers the systemic response, particularly if the initial signal originates at the leaf's rim. Either way, these aspects should be discussed in more detail as should be the role of systemically spreading electric and hydraulic waves alongside chemical signals in this context.

Minor points to address and comments:

l 41: herbivorous insects

l 51: Glutamate receptor-like (GLR) protein

l 77: catalysing the reaction between Glu and cysteine for generation...

l 118: remove "as"

l 161: "N-terminal amino-terminal domain" is redundant

l 178: "indicating that Glu-induced long-distance calcium wave is also dependent on GLR3.3" → replace indicating by confirming, as this was already shown previously, i.e., by Toyota et al. 2018 and Nguyen et al., 2018 which should be cited here

l 185: replace "Supplementary information, Fig. S7" with Extended Data Fig. 7

ll 228-229: "Our study found that the GSH-triggered systemic defense response was suppressed in GCaMP3/glr3.3 but nearly unaffected in GCaMP3/glr3.6. —> please specify that the calcium-mediated response was attenuated, as you did not monitor the hydraulic or electric response

l 338/339: replace 2th by 2nd

l 339: "larvae were first undergo 24 h of starvation" replace by first underwent 24h of starvation

l 340: "and then gently placed.." replace with "and were then gently placed..."

l 340: "were placed on plant" —> "placed on a plant"

Extended Data Figure 3a: enlarged inset is mirrored from original image for Col-0

Comment to: "It is surprising that such a high level of Glu failed to induce plant defense in pad2-1 (Extended Data Figs. 2 and 3), which seems to be inconsistent with the previously defined role of Glu as a critical wound signal that activates plant defense response."

The lack of a defense response in pad2-1 despite high glutamate levels could be attributed to several factors. The deficiency in GSH, along with reduced camalexin production and altered defense signaling, likely disrupts the balance of the plant's defense mechanisms. GSH is crucial for modulating both the salicylic acid (SA) and JA pathways. In pad2-1, insufficient GSH may downregulate the JA pathway, which is essential for defense activation, leading to a reduced JA response despite high glutamate levels. Additionally, the mutant's compromised GSH biosynthesis might shift defense responses towards the SA pathway, further diminishing the expected JA-mediated defense.

Version 1:

Reviewer comments:

Reviewer #1

(Remarks to the Author)

I congratulate the authors for successfully conducting new experiments that completely address my concerns about this impactful paper, which reports a new transmitter in systemic calcium-mediated defense signaling in plants. I am confident that this manuscript is now ready for publication.

Reviewer #2

(Remarks to the Author)

Reviewer #3

(Remarks to the Author)

The authors did a lot of additional experiments and addressed all my comments in satisfactory manner. They confirmed their initial hypothesis about the role of GSH in systemic Ca²⁺ signal propagation and I have no further comments. Very nice work!

Reviewer #4

(Remarks to the Author)

In the revised manuscript, Li et al. satisfactorily and thoroughly addressed all my major concerns and questions. Particularly, the manuscript was precisely rewritten and figures were amended to include pertinent suggestions (i.e., Extended Data Figs. 4, 6-8, 12, Suppl. Movies 6-8). The additional set of data supports and strengthens the findings that GSH (and not glutamate) is triggering plant defense signaling. I therefore recommend the paper in its current version for publication.

Dear Reviewers,

Thank you very much for your invaluable time and effort dedicated to reviewing our manuscript (NCOMMS-24-37210). We highly appreciate your insightful comments and constructive suggestions, which have immensely contributed to making our work more comprehensive and significant.

We have carefully studied your comments and successfully completed all the major experiments requested by you. We have also substantially revised our manuscript to fully address your questions.

Our newly obtained data are presented in Fig. 1e, 2c, 2h, 4c and Extended Data Figs. 2e, 4, 6, 7, 8, 12a and 12b in the revised manuscript. Some new data, falling into “helpful, but not essential” category, are just present in our response letter for your information. The major revisions of text in the manuscript are marked with blue color.

We would be grateful if our thoroughly revised manuscript will meet with your approval.

Our responses to the reviewers’ comments are listed as following.

☛ Reviewer #1's Questions and Our Responses

☛ Reviewer #2's Questions and Our Responses

☛ Reviewer #3's Questions and Our Response

☛ Reviewer #4's Questions and Our Response

Reviewer #1's Questions and Our Responses

Reviewer #1's Question 1 (R1Q1): Wounding signals are transmitted by certain molecules to trigger systemic calcium mediated defense signaling in plants. Previous studies identified glutamate (Glu) and beta-thioglucoside glucohydrolase (TGG) as these molecules. In this study, the authors propose glutathione (GSH) as a new transmitter. They analyzed *Arabidopsis thaliana pad2* mutant, which lacks gamma-glutamylcysteine synthetase activity that catalyzes Glu and cysteine for the subsequent generation of GSH. In the mutant, the Glu amount was doubled, and the GSH amount was reduced by 16 times. The authors found that amplitude and kinetics of cytosolic Ca^{2+} levels were reduced in the mutant, and transcript levels of defense response genes downstream of the Ca^{2+} change were diminished. Additionally, exogenous addition of GSH induced the Ca^{2+} wave and the induction of defense genes. Moreover, the Ca^{2+} wave induced by the exogenous GSH addition was diminished in the glutamate receptor *glr3.3* mutant. The finding of a new transmitter to regulate defense signaling with Ca^{2+} is influential for a wide community of researchers studying signal transduction, as the function of ligand-gated GLR channels in long-distance Ca^{2+} systemic signaling in plants remains unclear. The text is concise and clear, and the figures are of high quality. The mutant phenotypes and the effects of experimental treatments were well evaluated using multiple approaches, such as Ca^{2+} imaging, detection of marker gene expression, and quantification of metabolites with mass spectrometry. The effort and potential significance of the work presented are appreciated. However, there are some concerns regarding the identification of GSH as a transmitting molecule, and the authors are recommended to conduct additional experiments to verify their findings.

Response: Thank you very much for your enthusiastic appreciation of the novelty and quality of our work. We highly appreciate your insightful comments and constructive suggestions, which have significantly contributed to improving the quality of our work.

R1Q2: There is concern that the increase in H^+ , rather than GSH, might have induced the Ca^{2+} signaling. 0.05% (w/v) MES in the growth medium (pH 5.1) used to dissolve GSH approximately corresponds to 2.6 mM MES. When calculated using the Henderson-Hasselbalch equation with a pKa of 6.15 for MES, about 90% of 2.6 mM MES is protonated in the growth medium after adjustment to pH 5.1, and the remaining 0.26 mM MES is effective. When GSH is dissolved to 100 mM in the growth medium, 0.26 mM MES concentration is too low to prevent acidification against 100 mM GSH, and the pH value should have decreased to pH3 or less.

Response: Thank you for your valuable comments, which gives us the opportunity to

clarify the methodology. In our study, we dissolved GSH in a growth medium containing 1/2× MS salts, 1% (w/v) sucrose, and 0.05% (w/v) MES. The final pH of the solution was adjusted to 5.1, in accordance with methods described in previous studies (Science, 361: 1112). We apologize for the confusion caused by our description and have revised the pertinent sections of the manuscript to improve the clarity (lines 318-319), which is also listed here for your information:

Lines 318-319: GSH, Glu-Pro-Ala, L-Glu, and sorbitol were dissolved in growth medium [1/2× MS salts, 1% (w/v) sucrose and 0.05% (w/v) MES], which was then adjusted to pH 5.1.

To further investigate the influence of H⁺ on calcium signaling, we conducted additional experiments by applying a 100 mM sorbitol solution at pH 3 to the leaves of *GCaMP3* plants. Our results demonstrated that this acidic solution did not induce the long-distance transmission of calcium signals (please see the figure below). In contrast, notable Ca²⁺ signal transmission was observed in the systemic leaves following the application of a 100 mM GSH solution at pH 7 to leaf 1 of the *GCaMP3* plants (please see the figure below). These findings further reinforce our conclusion that GSH, rather than H⁺, is responsible for triggering long-distance calcium signaling in plants.

Application of GSH but not the acidic sorbitol solution could trigger long-distance transmission of calcium signal. [Ca²⁺]_{cyt} fluorescence signal imaging of *GCaMP3* plants after application of 100 mM sorbitol (pH 3) and 100 mM GSH (pH 7) to leaf 1 (L1) (white arrow, 0 s) respectively. White arrowheads (200 s, 400 s) indicate leaf 6 (L6), leaf 3 (L3) and leaf 4 (L4). Scale bars: 2 mm.

RIQ3: Further experiments are recommended to exclude the possibility that Glu, rather than GSH, induces the Ca^{2+} signal and defense gene expression. Even though Glu amounts globally increased in *pad2* mutant plant body, Glu may further increase locally and function to induce Ca^{2+} increase.

Response: Thank you for your suggestion. The data from *pad2-1* is able to validate the conclusions about the role of Glu or GSH in triggering plant systemic defense signaling. Glu was previously reported as a wound signal, triggering long-distance calcium-based defense signaling in Arabidopsis plants. According to this conclusion, Arabidopsis plants with higher levels of Glu would be expected to display increased resistance to insect attacks and pathogen infections. However, as shown in our study, we found that GSH-deficient plant mutant *pad2-1*, which has high accumulation of Glu, disharmoniously exhibited a series of reduced defense signaling events, such as wound-induced long-distance $[\text{Ca}^{2+}]_{\text{cyt}}$ transmission, systemic defense gene expression as well as plant resistance to pathogen infection and insect attack. We further showed that exogenous application of GSH could robustly trigger the fast and transient increase of $[\text{Ca}^{2+}]_{\text{cyt}}$ in systemic leaves and induce expression of a large number of key defense-responsive genes. These data strongly support the role of GSH as a signaling molecule to activate plant systemic defense signaling.

In light of your suggestion, we have carried out additional experiments to further exclude the possibility that Glu, rather than GSH, induces the Ca^{2+} signal and defense gene expression.

1) To rule out the possibility that GSH is cleaved by enzymes to release glutamate for Ca^{2+} signal propagation, we employed GGsTop, a well-known chemical inhibitor that specifically targets gamma-glutamyl transpeptidase (GGT), the enzyme responsible for converting GSH to Glu (Kamiyama et al., 2016). Our results showed that there were no significant differences in the Ca^{2+} signaling response when GSH was used in combination with GGsTop compared to that of GSH treatment alone (Extended Data Fig. 6 in the revised manuscript; please also see the figure shown below). These findings indicate that GSH is a signaling molecule that directly triggers long-distance transmission of $[\text{Ca}^{2+}]_{\text{cyt}}$ signals in plants.

Application of GGSTOP fails to suppress GSH-triggered systemic $[Ca^{2+}]_{cyt}$ transmission in plants. **a**, $[Ca^{2+}]_{cyt}$ fluorescence signal imaging of *GCaMP3* plants after application of 100 mM GSH combined with 0.5 mM GGsTOP and 0.5 mM GGsTOP alone to leaf 1 (L1) (white arrow, 0 s) respectively. White arrowheads (200 s, 400 s) indicate leaf 6 (L6), leaf 3 (L3) and leaf 4 (L4). Scale bars: 2 mm. **b**, Quantitative measurement of $[Ca^{2+}]_{cyt}$ levels in target leaf 6 at indicated times after application of 100 mM GSH combined with 0.5 mM GGsTOP and 0.5 mM GGsTOP alone to leaf 1 of *GCaMP3* plants. Data are mean \pm SD (n = 12 for GSH + GGsTOP, n = 7 for GGsTOP).

- 2) Moreover, we synthesized another tripeptide (Glu-Pro-Ala), which contains glutamate along with two different amino acids, and investigated the effect of this tripeptide in triggering plant systemic Ca^{2+} signal propagation. Our experiments revealed that tripeptide Glu-Pro-Ala failed to induce the long-distance transmission of Ca^{2+} signals in plants, which is quite different with the response observed with GSH (Extended Data Fig. 7 in the revised manuscript; please also see the figure attached below). These data reinforce that GSH-triggered systemic Ca^{2+} signal propagation is not caused by its conversion to glutamate.

Application of Glu-Pro-Ala to leaf 1 fails to induce long-distance $[Ca^{2+}]_{cyt}$ transmission in *GCaMP3* plants. **a**, $[Ca^{2+}]_{cyt}$ fluorescence signal imaging of *GCaMP3* plants after 100 mM Glu-Pro-Ala application (white arrow, 0 s) to leaf 1 (L1). White arrowhead (200 s) indicates leaf 6 (L6). Scale bar: 2 mm. **b**, Quantitative measurement of $[Ca^{2+}]_{cyt}$ levels in target leaf 6 at indicated times after application of 100 mM Glu-Pro-Ala to leaf 1 of *GCaMP3* plants. Data are mean \pm SD (n = 9). GSH-induced *GCaMP3* data from Figure 2c are reproduced (gray lines) to aid in comparison.

- 3) We further followed your suggestion to deny the possibilities that cells with high Glu accumulation may become less responsive to Glu-induced calcium signaling, further supporting our conclusion that the reduced defense signaling events in *pad2-1* was caused by GSH rather than Glu (please refer to our response to R1Q5 for details).

In summary, we have provided further evidence to support the role of GSH as a wound signal to trigger long-distance Ca^{2+} signaling in plants. These new data have significantly improved our manuscript. Thank you!

R1Q4: The authors speculate that GSH activates GLR3.3 to increase cytosolic Ca^{2+} based on previous reports on the binding of GSH to GLR3.3. However, to identify GSH as a transmitter to Ca^{2+} signal via GLR3.3, it is necessary to examine the activation of GLR3.3 by GSH *in vitro*.

Response: Thank you for your valuable suggestions. We have made extensive efforts to express and purify the GLR3.3 protein for *in vitro* experiments. However, we faced challenges in obtaining this protein due to its nature as a complex membrane protein, which complicates both its expression and purification. Our key finding is the identification of GSH as a wound signaling molecule that triggers long-distance transmission of $[Ca^{2+}]_{cyt}$ for activation of the defense response in plants. This conclusion is supported by plenty of *in vivo* and *in vitro* evidence and we have also followed your suggestions by performing additional experiments for further validation (e.g., R1Q2, R1Q3, R1Q5, R1Q6, R1Q9).

To establish that GSH relies on GLR3.3 to trigger systemic calcium-based defense signaling in plants, we generated *glr3.3* and *glr3.6* mutants within the *GCaMP3* background and applied GSH to the leaves of *GCaMP3/glr3.3* and *GCaMP3/glr3.6*. Notably, the GSH-induced systemic $[Ca^{2+}]_{cyt}$ transmission and defense gene expression was strongly reduced in the *GCaMP3/glr3.3* mutant, while it remained unaffected in

the *GCaMP3/glr3.6* mutant (Figure 4 in the revised manuscript). This provides ample evidence that GSH requires GLR3.3 to activate calcium-based defense signaling in plants. While it would be beneficial to further validate whether GSH can activate the GLR3.3 calcium channel *in vitro*, we believe that this is not essential for our main conclusion. Similar to our findings, previous studies on Glu (Science, 361: 1112) and TGG (Cell, 186: 1337), which genetically revealed small molecule-dependent calcium channels, did not further verify whether these small molecules could activate the channels *in vitro*. We would greatly appreciate your support in allowing us to investigate the activation of GLR3.3 by GSH in our future research.

In response to your comment, we have incorporated a discussion regarding the potential underlying mechanisms through which GSH may activate GLR3.3 in plants (lines 269-284), which is also shown below for your information.

Lines 269-284: The successful transmission of neural signals from one cell to another in animals relies on the mutual conversion of chemical signals (neurotransmitters) and electrical signals, where ion fluxes are induced by neurotransmitters and the release of neurotransmitters is dependent on electrical signals^{1,2,42}. In contrast, little is known about how signaling substances are transmitted in plants to activate long-distance signaling. Previous studies suggest that the dispersal of elicitors such as Glu may rely on bulk flow and diffusion¹⁴. Given that GSH has a molecular weight similar to that of amino acids, a plausible scenario is that when plants experience mechanical damage or insect attack, a large volume of GSH from the intracellular region may surge into the apoplastic region, inducing calcium influx by activation of the GLR3.3 ion channel. With the action of diffusion and bulk flow of GSH within the vascular bundles, calcium signals can be rapidly transmitted between leaves. In addition to the diffusion and bulk flow theory, other theories have also been proposed, such as the osmoelectric siphon model, which is based on membrane depolarization leading to cell water shedding into the apoplast followed by membrane repolarization and water uptake⁴³. More recently, osmotic pressure has been suggested to induce the systemic apoplastic release of signaling molecules like Glu¹⁵. Future studies aimed at uncovering the potential mechanisms underlying GSH-triggered systemic Ca²⁺ signal transmission would be fascinating.

R1Q5: line 107: The authors need to deny the possibility that cells are acclimated to increased Glu and becomes less sensitive to wound induced Glu, rather than the decrease of GSH.

Response: We appreciate your valuable suggestion and have conducted additional

experiments to deny the possibility that cells with high Glu accumulation may become less responsive to Glu-induced calcium signaling. We exogenously applied Glu to the leaves of *GCaMP3/pad2-1* plants, and found that Glu can still strongly trigger long-distance transmission of $[Ca^{2+}]_{cyt}$, with signal intensity comparable to that observed in *GCaMP3* plants (please see the figure below). These findings exclude the possibilities that cells with elevated Glu levels exhibit reduced sensitivity to Glu-triggered signaling.

Glu accumulation in *pad2-1* mutant does not attenuate Glu-triggered long-distance transmission of calcium signaling. **a**, $[Ca^{2+}]_{cyt}$ fluorescence signal imaging of *GCaMP3/pad2-1* plants after 100 mM Glu application to leaf 1 (L1) (white arrow, 0 s). White arrowheads (200 s, 400 s) indicate leaf 6 (L6), leaf 3 (L3) and leaf 4 (L4). Scale bars: 2 mm. **b**, Quantitative measurement of $[Ca^{2+}]_{cyt}$ levels in target leaf 6 at indicated times after application of 100 mM Glu to leaf 1 of *GCaMP3/pad2-1* plants. Data are mean \pm SD (n = 5).

R1Q6: line. 105: Ca^{2+} measurement of the *pad2* mutant with exogenous GSH may support the role of GSH.

Response: We followed your valuable suggestion to exogenously apply GSH on the leaf of *GCaMP3/pad2-1* and measured the Ca^{2+} signal. The results showed that GSH could effectively trigger long-distance transmission of $[Ca^{2+}]_{cyt}$ (please see the figure below). These data further support our conclusion that GSH acts as a wound signal to trigger plant systemic calcium-based defense signaling.

Application of GSH could trigger long-distance transmission of [Ca²⁺]_{cyt} signal in *pad2-1* mutant. **a**, [Ca²⁺]_{cyt} fluorescence signal imaging of *GCaMP3/pad2-1* plants following the application of 100 mM GSH to leaf 1 (L1) (white arrow, 0 s). White arrowheads (200 s, 400 s) indicate leaf 6 (L6), leaf 3 (L3) and leaf 4 (L4). Scale bars: 2 mm. **b**, Quantitative measurement of [Ca²⁺]_{cyt} levels in target leaf 6 at indicated times after application of 100 mM GSH to leaf 1 of *GCaMP3/pad2-1* plants. Data are mean ± SD (n = 7).

RIQ7: line 117 (Fig. 2c): Please explain the reason for the difference in graph shapes (3 clear peaks for Glu but 3 gradual peaks for GSH).

Response: Thank you for your insightful question regarding the differences in graph shapes for Ca²⁺ signal intensity between Glu and GSH. Variations in the number of biological replicates used to assess Ca²⁺ signal intensity contribute to the differences in graph shapes. Since previous studies have proved that Glu treatment could trigger long-distance calcium signaling (Science, 361: 1112), we therefore used Glu as a positive control with three replicates for data analysis in our original manuscript. In contrast, we employed nine biological replicates for the GSH treatment. To accommodate your comment, we have now increased the number of biological replicates for Glu treatment in Ca²⁺ signal analysis (Fig. 2c in the revised manuscript). The peak for Glu treatment becomes gradual as with the increasing number of biological replicates in Ca²⁺ signal analysis.

RIQ8: line 123: Results of JAZ10 in Fig. 1 and JAZ5 in Fig. 2h should be added for comparison to other figures.

Response: Thank you for your suggestion. All the 13 *JAZ* genes in Arabidopsis serve as responsive JA marker genes. Researchers normally select a subset of *JAZs* to assess the expression levels, thereby indicating JA signal intensity. In Figure 1, we examined

the expression of JA biosynthesis gene *OPR3* along with two JAZ genes (*JAZ5* and *JAZ7*). We have now included the expression data for *JAZ7* in the revised manuscript. For Figure 2, we have followed your suggestion to update the data and included the results of *JAZ5*.

R1Q9: line 143: For comparison with Fig. 1 and 2, qRT-PCR of *OPR3*, *JAZ5*, and *JAZ10* should be added.

Response: We have followed your suggestion to add the qRT-PCR results of *OPR3*, *JAZ5*, *JAZ7*, and *JAZ10* in our revised manuscript (Extended Fig. 8 in the revised manuscript) and also attached below for your information. Thank you!

Relative gene expression level of JA-responsive genes in the target leaf 6 one hour after application of 100 mM GSH to leaf 1 of WT plants. Data are mean \pm SD (n=3). **P < 0.01, *P < 0.001, Welch's t test.**

R1Q10: line 172: *JAZ5* and *JAZ10* should be analyzed in concordance with other experiments in Figs. 1 and 2.

Response: We have followed your suggestion to update the data and included the results of *JAZ5* and *JAZ10* in the revised manuscript (Figure 4c in the revised manuscript). Thank you!

R1Q11: Figs. 1h, 2g, and 4b: ROIs should be indicated.

Response: We have followed your suggestion to add a diagram to mark the ROI (Extended data Fig. 4 in the revised manuscript), which is also attached below for your information. Thank you!

Diagram showing the regions of interest (ROI) used to assess the $[Ca^{2+}]_{cyt}$ increase in systemic leaf 6 after wounding or application of chemical molecules to leaf 1. The entire leaf 6 was used to analyze the $[Ca^{2+}]_{cyt}$ increase.

R1Q12: The authors should use Welch's t-test instead of the Student's t-test. The Student's t-test assumes equal variance of the two populations compared, but Welch's t-test does not. It is usually unclear whether the two populations compared have equal variance, so Welch's t-test is more reliable.

Response: We have followed your suggestion to use Welch's t-test instead of the Student's t-test to analyze the data in the revised manuscript. Thank you!

R1Q13: For multiple comparisons, the authors sequentially use one-way ANOVA and then Dunnett's test. Although such a sequential application of ANOVA and a multiple comparison method is often used, this is logically flawed [Ruxton and Beauchamp (2008) Behavioral Ecology, 19: 690–693]. It is better to simply use Dunnett's test without the prior use of ANOVA.

Response: We have followed your suggestion to simply use Dunnett's test for multiple comparisons. Thank you!

R1Q14: Fig. 1j and Extended Fig. 2e lack statistical tests. For Extended Fig. 2e, Steel's test (non-parametric version of Dunnett's test) can be applicable. For Fig. 1j, Fisher's exact test is one possible choice.

Response: We have now included the appropriate statistical tests in the revised manuscript. Specifically, we applied Fisher's exact test for Fig. 1i and the Kruskal-Wallis test for Extended Fig. 2e. We appreciate your valuable suggestions!

Reviewer #2's Questions and Our Responses

Reviewer #2's Question 1 (R2Q1): I co-reviewed this manuscript with one of the reviewers who provided the listed reports. This is part of the Nature Communications initiative to facilitate training in peer review and to provide appropriate recognition for Early Career Researchers who co-review manuscripts.

Response: We sincerely appreciate the time and effort you dedicated to reviewing our manuscript. Your insightful comments and constructive suggestions have significantly enhanced the quality of our work, and we are grateful for your valuable feedback.

Reviewer #3's Questions and Our Responses

Reviewer #3's Question 1 (R3Q1): The manuscript entitled: Glutathione triggers leaf-to-leaf, calcium-based plant defense signalling submitted by Li et al. to Nature Communications investigates the role of glutathione (GSH) in plant systemic signalling. In their experiments, they use series of elegant genetic and pharmacological approaches to show that GSH is an important component of wound induced Ca^{2+} systemic signalling in plants. So far, this function was ascribed mainly to amino acid L-glutamate and so called Ricca factor, but it has been clear that systemic wound signal is much more complex. Using Arabidopsis intracellular calcium $[\text{Ca}^{2+}]_{\text{cyt}}$ reporter lines and their crosses with *glr* and *pad2-1* mutants, they elegantly demonstrated propagation of $[\text{Ca}^{2+}]_{\text{cyt}}$ signal. Further, they showed that glutathione Ca^{2+} signalling response is dependent on GLR3.3 receptor, similar to L-Glu. Using molecular docking, they suggested a possible binding pocket in ATD subunit of GLR3.3. The study seems to be carefully done with presented experimental setup (but see my comments below), with sufficient repetition and the manuscript is clearly and concisely written.

Response: Many thanks for your appreciation of the novelty and significance of our work. Your insightful comments have greatly improved the quality of our work.

R3Q2: Glutathione (GSH) is tripeptide composed of glutamate, glycine and cysteine. You suggested its specific role in systemic Ca^{2+} signal propagation. However, is it possible that after wounding, when cellular content an enzyme compartmentalization is disrupted, the GSH is cleaved and the effect is mediated through released glutamate? Is the effect of GSH indeed a separate phenomenon? GSH induced exactly the same response as L-Glu and both are dependent on GLR3.3. Maybe, more appropriate negative control than osmotic control sorbitol, would be another tripeptide containing glutamate but two other amino acids.

Response: We appreciate your suggestion to investigate whether the effects of GSH could be mediated by cleaved glutamate. We have followed your suggestion to synthesize an additional tripeptide (Glu-Pro-Ala), which contains glutamate along with two different amino acids, and investigated the effect of this tripeptide in triggering plant systemic Ca^{2+} signal propagation. Our experiments revealed that tripeptide Glu-Pro-Ala failed to induce the long-distance transmission of Ca^{2+} signals in plants, which is quite different with the response observed with GSH treatment (Extended Data Fig. 7 in the revised manuscript; please also see the figure attached below). These data suggest that GSH-triggered systemic Ca^{2+} signal propagation is not caused by the conversion of GSH to glutamate.

Application of Glu-Pro-Ala to leaf 1 fails to induce long-distance [Ca²⁺]_{cyt} transmission in *GCaMP3* plants. a, [Ca²⁺]_{cyt} fluorescence signal imaging of *GCaMP3* plants after 100 mM Glu-Pro-Ala application (white arrow, 0 s) to leaf 1 (L1). White arrowhead (200 s) indicates leaf 6 (L6). Scale bar: 2 mm. **b**, Quantitative measurement of [Ca²⁺]_{cyt} levels in target leaf 6 at indicated times after application of 100 mM Glu-Pro-Ala to leaf 1 of *GCaMP3* plants. Data are mean ± SD (n = 9). GSH-induced *GCaMP3* data from Figure 2c are reproduced (gray lines) to aid in comparison.

To further rule out the possibility that GSH is cleaved by enzymes to release glutamate for Ca²⁺ signal propagation, we employed GGsTop, a well-known chemical inhibitor that specifically targets gamma-glutamyl transpeptidase (GGT), the enzyme responsible for converting GSH to Glu (Kamiyama et al., 2016). Our results showed that there were no significant differences in the Ca²⁺ signaling response when GSH was used in combination with GGsTop compared to that of GSH treatment alone (Extended Data Fig. 6 in the revised manuscript; please see the figure shown below). These findings provide additional evidence to reinforce GSH as a signaling molecule that directly triggers long-distance transmission of [Ca²⁺]_{cyt} signals in plants.

Application of GGsTOP fails to suppress GSH-triggered systemic $[Ca^{2+}]_{cyt}$ transmission in plant. a, $[Ca^{2+}]_{cyt}$ fluorescence signal imaging of *GCaMP3* plants after application of 100 mM GSH combined with 0.5 mM GGsTOP and 0.5 mM GGsTOP alone to leaf 1 (L1) (white arrow, 0 s) respectively. White arrowheads (200 s, 400 s) indicate leaf 6 (L6), leaf 3 (L3) and leaf 4 (L4). Scale bars: 2 mm. **b,** Quantitative measurement of $[Ca^{2+}]_{cyt}$ levels in target leaf 6 at indicated times after application of 100 mM GSH combined with 0.5 mM GGsTOP and 0.5 mM GGsTOP alone to leaf 1 of *GCaMP3* plants. Data are mean \pm SD (n = 12 for GSH + GGsTOP, n = 7 for GGsTOP).

R3Q3: Transcriptomic analysis (Fig. 3). You applied GSH and L-Glu after 30 min recovery period after leaf pre-cutting. Then, transcriptomic analyses were done 1 hour after application of GSH, L-Glu and wounding and compared to mock control. Does the mock control contain this pre-cutting stimulus? Otherwise, increased expression of defence – related genes in L-Glu and GSH treated plants may be the result of pre-cutting stimulus (analysed 1.5 h after). Recently, Bellandi et al. 2022 (Sci. Adv.) criticized this pre-cutting approach and found that glutamate alone without wounding component cannot induce JA response in systemic leaves. In your movies, it is clearly visible wound after pre-cut treatment and I cannot believe that this intervention does not affect defence-gene expression after 1.5 hours.

Response: Thank you for your valuable comments, which give us the opportunity to clarify the misunderstanding. In our study, we followed the methods described in previous study (Science, 361: 1112) to observe small molecules (such as GSH and Glu)-induced systemic Ca^{2+} propagation: The small molecules (such as GSH and Glu) were applied to the wound site of plant leaves after a 30-minute recovery period after pre-cutting. This timing coincides with the period when the wound-induced calcium response has stabilized and returned to the baseline levels, ensuring that the observed effects of the applied small molecules are attributable to their specific actions rather than residual responses from the initial wounding.

In contrast to calcium signaling, the downstream jasmonate signaling is much slower, taking up to at least three hours for the expression of wound-induced JA-responsive genes to return to the baseline levels (Science, 361: 1112). Therefore, for transcriptomic analysis and gene expression measurement, the small molecules (such as GSH and Glu) were applied to the wound site of plant leaves after a 5-hour recovery period after pre-cutting, with systemic leaves harvested after this 5-hour recovery period serving as mock controls. This timing ensures that the effects on gene expression can be accurately

attributed to GSH and Glu, rather than the wounding treatment.

We apologize for any misunderstanding caused by our presentation and have revised the methodology in the manuscript to enhance clarity (lines 319-323). This revised methodology is also summarized here for your information:

Lines 319-323: 5 μL of the solution was applied to the cut edge of leaf 1 after 30 minutes recovery period after pre-cutting for assessment of small molecules-induced Ca^{2+} signal transmission. In contrast, a longer recovery period of 5 hours post pre-cutting was required for the assessment of small molecules-activated gene expression levels.

R3Q4: Did you analysed if your data have equal variances and normal distribution? Equal variances and normality are assumptions for t-test and ANOVA.

Response: We did assess the equal variances and normal distributions of our data before employing t-tests and ANOVA in our original manuscript. However, we have made modifications to the statistical methods in the revised version according Reviewer #1 and #2's suggestions. We replaced the Student's t-test with Welch's t-test, and removed ANOVA analysis. This adjustment means that the statistical analyses no longer require the assumptions of equal variances and normal distribution of the data. Thank you!

R3Q5: Page 5/line107: „Taken together, the above-mentioned data suggest that GSH might serve as an indispensable wound signal that triggers wound-induced systemic $[\text{Ca}^{2+}]_{\text{cyt}}$ transmission, activates JA biosynthesis, and regulates plant defense responses. “ Because the Ca^{2+} signalling and defence response is only attenuated but not completely blocked in *pad2-1* mutant, I found this not as indispensable but maybe redundantly dependent. GSH is probably only modulator. Except its role in Ca^{2+} signalling you showed, it is known that GSH can modulate phytohormone response dependent on repressor stabilization/degradation (e.g. IAA and JA signalling), and can act as IAA or JA signalling enhancer, Pasternak et al., 2020). So the increased susceptibility of *pad2-1* mutant plants to herbivore damage can be connected also with other mechanisms of GSH action.

Response: We fully agree with you that the susceptibility of *pad2-1* to insect attack was not only ascribed to the reduced Ca^{2+} signaling and JA biosynthesis. Other factors as you listed may also contribute to the increased susceptibility of *pad2-1* mutant plants to herbivore damage. We have followed your suggestion to replace the word “indispensable” by “important” to temper the tone of the statement in the revised

manuscript. Thank you!

R3Q6: Maybe, you should discuss how the GSH may bind and activate GLR3.3 receptor. A possible scenario is that GSH, as the main transport form of sulphur in phloem, is released upon wounding?

Response: In light of suggestion, we have added discussion on how the GSH may bind and activate GLR3.3 receptor (lines 269-284), which is also shown below for your information. Thank you!

Lines 269-284: The successful transmission of neural signals from one cell to another in animals relies on the mutual conversion of chemical signals (neurotransmitters) and electrical signals, where ion fluxes are induced by neurotransmitters and the release of neurotransmitters is dependent on electrical signals^{1,2,42}. In contrast, little is known about how signaling substances are transmitted in plants to activate long-distance signaling. Previous studies suggest that the dispersal of elicitors such as Glu may rely on bulk flow and diffusion¹⁴. Given that GSH has a molecular weight similar to that of amino acids, a plausible scenario is that when plants experience mechanical damage or insect attack, a large volume of GSH from the intracellular region may surge into the apoplastic region, inducing calcium influx by activation of the GLR3.3 ion channel. With the action of diffusion and bulk flow of GSH within the vascular bundles, calcium signals can be rapidly transmitted between leaves. In addition to the diffusion and bulk flow theory, other theories have also been proposed, such as the osmoelectric siphon model, which is based on membrane depolarization leading to cell water shedding into the apoplast followed by membrane repolarization and water uptake⁴³. More recently, osmotic pressure has been suggested to induce the systemic apoplastic release of signaling molecules like Glu¹⁵. Future studies aimed at uncovering the potential mechanisms underlying GSH-triggered systemic Ca²⁺ signal transmission would be fascinating.

Reviewer #4's Questions and Our Responses

Reviewer #4's Question 1 (R4Q1): Li et al. performed a hairpin RNA-based screen in Arabidopsis and identified a glutathione (GSH)-deficient mutant in the *At4g23100* gene (gamma-glutamylcysteine synthetase, also known as *pad2-1*). This mutant exhibited a 1.8-fold increase in glutamate levels and a 17-fold decrease in GSH compared to wild-type plants. Hormone assays revealed that *pad2-1* mutants had reduced JA-derivatives and lower expression of JA-induced genes upon wounding. Using GCaMP3-based calcium imaging, they observed a diminished Ca^{2+} response to wounding in the mutant. When GSH was applied to GCaMP3 plants, it induced dose-dependent, LaCl_3 -sensitive, systemic Ca^{2+} signals and upregulated defense-related genes. Further gene expression analysis highlighted specific clusters of genes differentially regulated by GSH versus glutamate or wounding, many associated with plant defense, JA response, and wounding. Molecular docking studies suggested that GSH might bind to GLR3.3's ATD. Based on their data the authors propose that GSH could serve as a wound signal, initiating systemic Ca^{2+} signaling, JA biosynthesis, and defense responses. The manuscript is well-written, and the experiments are robust, carefully conducted, and largely presented effectively. The identification of GSH as a potential signaling molecule in systemic wound signaling is a significant discovery with considerable implications for the field. However, the analysis of certain datasets, particularly those related to calcium signaling, along with their discussion and interpretation, require further refinement. A functional model integrating the findings with current literature would be desirable.

Response: Thank you very much for your enthusiastic appreciation on the significance and novelty of our study. Your insightful comments and constructive suggestions have significantly improved the quality of our work

R4Q2: The authors should explicitly state that the GSH concentration used in their experiments is several orders of magnitude higher than the typical cellular concentrations, which reportedly range from 0.01 to 15 mM. While local GSH application (100 mM) to a leaf triggers a systemic calcium response, the kinetics and amplitude of this response differ from those triggered by glutamate. Notably, the GSH-induced systemic Ca^{2+} signal appears to originate at the leaf margin, potentially in secondary or tertiary veins, and propagates inward toward the center. This contrasts with the expected signal pattern from wounding or glutamate application, where the Ca^{2+} wave typically starts at the petiole or main vasculature and moves outward toward the leaf rim (compare Supplemental Movies 3 and 2). Was this pattern consistently

observed? If so, this phenomenon should be emphasized and discussed, as it deviates from previously reported Ca^{2+} signaling patterns induced by glutamate or wounding. If the pattern holds true, it could indicate a spatially distinct response mechanism for GSH. Given these potential spatial differences in calcium signal propagation, it's crucial to ensure that the selection and positioning of the ROIs are comparable for consistent data analysis. Was a specific leaf area or the petiole selected for analysis, or was the entire leaf considered? Please include this methodological detail in the methods section, as it is currently missing.

Response: We appreciate your insightful suggestions, which have significantly improved our work. In our revised manuscript, we have followed your suggestion to clarify the GSH concentrations, discuss the distinct calcium signal propagation pattern, and provide detailed methodology on the selection and analysis of the ROI.

1) **Clarification of GSH Concentration:** As the major non-protein thiol, GSH is abundant in plant tissues, with the highest concentrations reaching up to 15 mM, as you mentioned. In addition to applying a concentration of 100 mM GSH externally, we also applied GSH with gradient concentrations to the leaf of plants. We found that $[\text{Ca}^{2+}]_{\text{cyt}}$ elevation was rapidly detected in the unwounded distal leaves after the application of GSH, even at a concentration of 5 mM, which falls within its physiological range in plant tissues. Moreover, more systemic leaves showed $[\text{Ca}^{2+}]_{\text{cyt}}$ signal changes with the increased GSH concentrations (5 mM, 10 mM, 25 mM, 50 mM, 100 mM) (Extended Data Fig. 5 in the revised manuscript). In light of your suggestion, we have elaborated on the concentrations of GSH that were externally applied to the wounds of leaf tissues and explicitly stated the relationship between exogenous concentrations and physiological concentrations of GSH in the revised manuscript (lines: 323-325), which is also shown below for your information.

Lines 323-325: GSH at concentrations of 0.01 mM, 0.1 mM, 1 mM, 5 mM, 10 mM, 25 mM, 50 mM, and 100 mM, which are within or exceeding its physiological concentrations, were selected for analyzing GSH-triggered systemic calcium-based defense signaling *in vitro*.

2) **Discussion of Calcium Signal Pattern:** During our experiments, we also noticed that GSH and Glu induced distinct $[\text{Ca}^{2+}]_{\text{cyt}}$ transmission patterns in systemic leaves. The spread of systemic $[\text{Ca}^{2+}]_{\text{cyt}}$ triggered by GSH preferentially initiated from marginal veins and rapidly transmitted to the mid veins, whereas Glu mediated $[\text{Ca}^{2+}]_{\text{cyt}}$ transmission mainly propagated from mid veins to marginal veins (Extended Data Fig. 12, Supplementary Movie 6 and 7 in the revised manuscript)

(please also see the figure below). Re-examination of the data shown in the previous study revealing Glu as a wound signal (Science, 361: 1112) also supports our observation. Intriguingly, the transmission pattern of wounding-induced $[Ca^{2+}]_{cyt}$ contained both directions of the mid-to-marginal veins and the marginal-to-mid veins (Extended Data Fig. 12 and Supplementary Movie 8 in the revised manuscript) (please also see the figure below). Elucidation of the underlying mechanisms and biological significance of different $[Ca^{2+}]_{cyt}$ transmission patterns induced by GSH and Glu would help us to further understand the highly sophisticated systemic defense systems against insect attack. We have followed your suggestion to add this discussion of Ca^{2+} pattern in the revised manuscript (lines 224-236).

Distinct calcium transmission pattern in plant systemic leaves was triggered by GSH and Glu. a, The representative images of early, middle, and later stages of GSH-, Glu- and wound-triggered $[Ca^{2+}]_{cyt}$ transmission in leaf 6 (L6) of *GCaMP3* plants. Note that GSH triggers the systemic $[Ca^{2+}]_{cyt}$ propagation in the marginal-to-mid vein direction (refers to Supplementary Movie 6 for details) while Glu induces the mid-to-marginal vein $[Ca^{2+}]_{cyt}$ transmission (refers to Supplementary Movie 7 for details). Wound-induced calcium transmission (refers to Supplementary

Movie 8 for details). Scale bars: 1 mm. **b**, The schematic diagrams of transmission directions of $[Ca^{2+}]_{cyt}$ triggered by Glu (left), GSH (middle), or wounding (right) in systemic leaves.

- 3) **Methodological Details on ROI Selection:** The entire leaf was selected for analyzing Ca^{2+} increases in our study. We have added a diagram showing the ROI (Extended Data Fig. 4 in the revised manuscript; please also see the figure attached below) and included the detail ROI in the methodology based on your suggestion (line 316), which is also shown below for your information.

Diagram showing the regions of interest (ROI) used to assess the $[Ca^{2+}]_{cyt}$ increase in systemic leaf 6 after wounding or application of chemical molecules to leaf 1. The entire leaf 6 was used to analyze the $[Ca^{2+}]_{cyt}$ increase.

Line 316: The entire leaf area was selected as ROI for Ca^{2+} signal analysis.

R4Q3: Was the effect of GSH on the $[Ca^{2+}]_{cyt}$ also analysed in the *GCaMP3/pad2-1* background? If so, how was the calcium signal compared to the other lines?

Response: In light of your suggestion, we have analyzed the effect of GSH-triggered $[Ca^{2+}]_{cyt}$ in *GCaMP3/pad2-1* plants. Our results showed that GSH could effectively trigger long-distance transmission of $[Ca^{2+}]_{cyt}$ (please see the figure below).

Application of GSH triggers long-distance transmission of $[Ca^{2+}]_{cyt}$ signal. **a**, $[Ca^{2+}]_{cyt}$ fluorescence signal imaging of *GCaMP3/pad2-1* plants following the

application of 100 mM GSH to leaf 1 (L1) (white arrow, 0 s). White arrowheads (200 s, 400 s) indicate leaf 6 (L6), leaf 3 (L3) and leaf 4 (L4). Scale bars: 2 mm. **b**, Quantitative measurement of $[Ca^{2+}]_{\text{cyt}}$ levels in target leaf 6 at indicated times after application of 100 mM GSH to leaf 1 of *GCaMP3/pad2-1* plants. Data are mean \pm SD (n = 7).

R4Q4: The authors should discuss the potential mechanisms underlying GSH-mediated signaling in their analysis and maybe summarise this in a functional model. Given that the largest GSH reserves are intracellular, for GSH to interact with the ATD of GLR3.3s, which are presumed to function on the plasma membrane, it would first need to reach the apoplasm. The authors should comment whether they believe that GSH itself is transported from the wounded site to systemic leaves? If this is the case, how do they reconcile this with the presumably rapid breakdown (within minutes) of GSH in the apoplasm, i.e., by enzymes like gamma-glutamyl transpeptidase, which would convert GSH into glutamate? This raises the question of whether glutamate could actually be the "bioactive" signal responsible for the observed effects, at least partially. To test this hypothesis, the authors could (1) employ a genetically encoded redox sensor like roGFP2 to monitor changes in redox state or, as the function of this sensor might be affected in the apoplasm, (2) compare the effects of GSH and glutamate application on plants expressing a genetically encoded glutamate sensor, such as GluSnFR or FLIPE. Since these sensors specifically detect glutamate, an increase in GSH concentration should not affect fluorescence, allowing for a clearer distinction between the roles of GSH and glutamate in signaling. If the authors suggest that GSH is immobile, they should address what triggers the systemic response, particularly if the initial signal originates at the leaf's rim. Either way, these aspects should be discussed in more detail as should be the role of systemically spreading electric and hydraulic waves alongside chemical signals in this context.

Response: Thank you for your insightful suggestions. We have carried out additional experiments to exclude the possibilities that GSH is converted into Glu to trigger systemic calcium-based defense response, and discussed the potential mechanisms underlying GSH-mediated signaling in the revised manuscript.

- 1) To rule out the possibility that GSH is cleaved by enzymes to release glutamate for Ca^{2+} signal propagation, we employed GGsTop, a well-known chemical inhibitor that specifically targets gamma-glutamyl transpeptidase (GGT), the enzyme responsible for converting GSH to Glu (Kamiyama et al., 2016). Our results showed that there were no significant differences in the Ca^{2+} signaling response when GSH

was used in combination with GGsTop compared to that of GSH treatment alone (Extended Data Fig. 6 in the revised manuscript; please also see the figure shown below). These findings indicate that GSH is a signaling molecule that directly triggers long-distance transmission of $[Ca^{2+}]_{\text{cyt}}$ signals in plants.

Application of GGsTOP fails to suppress GSH-triggered systemic $[Ca^{2+}]_{\text{cyt}}$ transmission in plant. **a**, $[Ca^{2+}]_{\text{cyt}}$ fluorescence signal imaging of *GCaMP3* plants after application of 100 mM GSH combined with 0.5 mM GGsTOP and 0.5 mM GGsTOP alone to leaf 1 (L1) (white arrow, 0 s) respectively. White arrowheads (200 s, 400 s) indicate leaf 6 (L6), leaf 3 (L3) and leaf 4 (L4). Scale bars: 2 mm. **b**, Quantitative measurement of $[Ca^{2+}]_{\text{cyt}}$ levels in target leaf 6 at indicated times after application of 100 mM GSH combined with 0.5 mM GGsTOP and 0.5 mM GGsTOP alone to leaf 1 of *GCaMP3* plants. Data are mean \pm SD (n = 12 for GSH + GGsTOP, n = 7 for GGsTOP).

Moreover, we synthesized an additional tripeptide (Glu-Pro-Ala), which contains glutamate along with two different amino acids, and investigated the effect of this tripeptide in triggering plant systemic Ca^{2+} signal propagation. Our experiments revealed that tripeptide Glu-Pro-Ala failed to induce the long-distance transmission of Ca^{2+} signals in plants, which is quite different with the response observed with GSH (Extended Data Fig. 7 in the revised manuscript; please also see the figure attached below). These data reinforce that GSH-triggered systemic Ca^{2+} signal propagation is not caused by the conversion of GSH to glutamate.

Application of Glu-Pro-Ala to leaf 1 fails to induce long-distance $[Ca^{2+}]_{cyt}$ transmission in *GCaMP3* plants. **a**, $[Ca^{2+}]_{cyt}$ fluorescence signal imaging of *GCaMP3* plants after 100 mM Glu-Pro-Ala application (white arrow, 0 s) to leaf 1 (L1). White arrowhead (200 s) indicates leaf 6 (L6). Scale bar: 2 mm. **b**, Quantitative measurement of $[Ca^{2+}]_{cyt}$ levels in target leaf 6 at indicated times after application of 100 mM Glu-Pro-Ala to leaf 1 of *GCaMP3* plants. Data are mean \pm SD (n = 9). GSH-induced *GCaMP3* data from Figure 2c are reproduced (gray lines) to aid in comparison.

- 2) Currently, several theories have been proposed regarding the mechanisms behind small molecule-triggered systemic calcium defense responses in plants, including diffusion, bulk flow, and the osmoelectric siphon model. GSH may activate signaling responses through one or more of these pathways, an aspect that needs to be clarified in future studies. In our revised manuscript, we have included an in-depth discussion on the mechanisms of GSH action based on your suggestions (lines 269-284), which is also provided below for your reference.

Lines 269-284: The successful transmission of neural signals from one cell to another in animals relies on the mutual conversion of chemical signals (neurotransmitters) and electrical signals, where ion fluxes are induced by neurotransmitters and the release of neurotransmitters is dependent on electrical signals^{1,2,42}. In contrast, little is known about how signaling substances are transmitted in plants to activate long-distance signaling. Previous studies suggest that the dispersal of elicitors such as Glu may rely on bulk flow and diffusion¹⁴. Given that GSH has a molecular weight similar to that of amino acids, a plausible scenario is that when plants experience mechanical damage or insect attack, a large volume of GSH from the intracellular region may surge into the apoplastic region, inducing calcium influx by activation of the GLR3.3 ion channel. With the action of diffusion and bulk flow of GSH within the vascular bundles, calcium signals can be rapidly transmitted between leaves. In addition to the diffusion and bulk flow

theory, other theories have also been proposed, such as the osmoelectric siphon model, which is based on membrane depolarization leading to cell water shedding into the apoplast followed by membrane repolarization and water uptake⁴³. More recently, osmotic pressure has been suggested to induce the systemic apoplastic release of signaling molecules like Glu¹⁵. Future studies aimed at uncovering the potential mechanisms underlying GSH-triggered systemic Ca²⁺ signal transmission would be fascinating.

R4Q5: 141: herbivorous insects

Response: We have made the correction. Thank you!

R4Q6: 151: Glutamate receptor-like (GLR) protein

Response: We have corrected the words according to your suggestion. Thank you!

R4Q7: 77: catalysing the reaction between Glu and cysteine for generation...

Response: We have followed your suggestion to make the correction. Thank you!

R4Q8: 118: remove “as”

Response: We have removed the word “as” according to your suggestion. Thank you!

R4Q9: 1161: “N-terminal amino-terminal domain” is redundant

Response: We have corrected “N-terminal amino-terminal domain” to “amino-terminal domain”. Thank you!

R4Q10: 178: “indicating that Glu-induced long-distance calcium wave is also dependent on GLR3.3” —> replace indicating by confirming, as this was already shown previously, i.e., by Toyota et al. 2018 and Nguyen et al., 2018 which should be cited here.

Response: We have followed your suggestion to replace ‘indicating’ by ‘confirming’ and also cited the mentioned literatures. Thank you!

R4Q11: 185: replace “Supplementary information, Fig. S7” with Extended Data Fig. 7

Response: We have made the correction. Thank you!

R4Q12: 228-229: “Our study found that the GSH-triggered systemic defense response was suppressed in GCaMP3/glr3.3 but nearly unaffected in GCaMP3/glr3.6. —> please specify that the calcium-mediated response was attenuated, as you did not monitor the hydraulic or electric response.

Response: We have followed your suggestion to make the correction. Thank you!

R4Q13: 338/339: replace 2th by 2nd

Response: We have made the correction. Thank you!

R4Q14: 339: “larvae were first undergo 24 h of starvation” replace by first underwent 24h of starvation.

Response: We have made the correction. Thank you!

R4Q15: 340: “and then gently placed..” replace with “and were then gently placed...”

Response: We have followed your suggestion to make the correction. Thank you!

R4Q16: l 340: “were placed on plant” —> “placed on a plant”

Response: We have made the correction. Thank you!

R4Q17: Extended Data Figure 3a: enlarged inset is mirrored from original image for Col-0.

Response: We have now followed your suggestion to display the original image for Col-0 in the revised manuscript. Thank you!

R4Q18: Comment to: “It is surprising that such a high level of Glu failed to induce

plant defense in *pad2-1* (Extended Data Figs. 2 and 3), which seems to be inconsistent with the previously defined role of Glu as a critical wound signal that activates plant defense response.” The lack of a defense response in *pad2-1* despite high glutamate levels could be attributed to several factors. The deficiency in GSH, along with reduced camalexin production and altered defense signaling, likely disrupts the balance of the plant’s defense mechanisms. GSH is crucial for modulating both the salicylic acid (SA) and JA pathways. In *pad2-1*, insufficient GSH may downregulate the JA pathway, which is essential for defense activation, leading to a reduced JA response despite high glutamate levels. Additionally, the mutant’s compromised GSH biosynthesis might shift defense responses towards the SA pathway, further diminishing the expected JA-mediated defense.

Response: Thank you for your valuable comment. We fully agree with you that in addition to the GSH-triggered calcium-based defense signaling shown in our study, several other factors as you listed in your comment, such as the reduced camalexin production, the crosstalk among GSH, SA, and JA signals might also contribute to the reduced defense signaling of *pad2-1* mutant plant. In light of your comment, we have revised the conclusion sentence of this section to temper the tone of the statement by replacing the word “indispensable” with “important” (lines 112-114 in the revised manuscript), which is also shown below for your information.

The original statement: “Taken together, the above-mentioned data suggest that GSH might serve as an indispensable wound signal that triggers wound-induced systemic $[Ca^{2+}]_{cyt}$ transmission, activates JA biosynthesis, and regulates plant defense responses.”

The revised statement: “Taken together, the above-mentioned data suggest that GSH might serve as an **important** wound signal that triggers wound-induced systemic $[Ca^{2+}]_{cyt}$ transmission, activates JA biosynthesis, and regulates plant defense responses.”

Responses to reviewers' comments

Reviewer #1 (Remarks to the Author):

I congratulate the authors for successfully conducting new experiments that completely address my concerns about this impactful paper, which reports a new transmitter in systemic calcium-mediated defense signaling in plants. I am confident that this manuscript is now ready for publication.

Response: Thank you very much for your encouraging feedback. We greatly appreciate your time in carefully reviewing our manuscript.

Reviewer #2 (Remarks to the Author):

Response: Thank you very much for your encouraging feedback. We greatly appreciate your time in carefully reviewing our manuscript.

Reviewer #3 (Remarks to the Author):

The authors did a lot of additional experiments and addressed all my comments in satisfactory manner. They confirmed their initial hypothesis about the role of GSH in systemic Ca^{2+} signal propagation and I have no further comments. Very nice work!

Response: Thank you very much for your encouraging feedback. We greatly appreciate your time in carefully reviewing our manuscript.

Reviewer #4 (Remarks to the Author):

In the revised manuscript, Li et al. satisfactorily and thoroughly addressed all my major concerns and questions. Particularly, the manuscript was precisely rewritten and figures were amended to include pertinent suggestions (i.e., Extended Data Figs. 4, 6-8, 12, Suppl. Movies 6-8). The additional set of data supports and strengthens the findings that GSH (and not glutamate) is triggering plant defense signaling. I therefore recommend the paper in its current version for publication.

Response: Thank you very much for your encouraging feedback. We greatly appreciate your time in carefully reviewing our manuscript.